# Splitting Steepest Descent for Growing Neural Architectures

Qiang Liu
UT Austin
lqiang@cs.utexas.edu

Lemeng Wu [*]
UT Austin
lmwu@cs.utexas.edu

Dilin Wang [*]
UT Austin
dilin@cs.utexas.edu

## Abstract

We develop a progressive training approach for neural networks which adaptively grows the network structure by splitting existing neurons to multiple off-springs. By leveraging a functional steepest descent idea, we derive a simple criterion for deciding the best subset of neurons to split and a *splitting gradient* for optimally updating the off-springs. Theoretically, our splitting strategy is a second-order functional steepest descent for escaping saddle points in an $\infty$-Wasserstein metric space, on which the standard parametric gradient descent is a first-order steepest descent. Our method provides a new practical approach for optimizing neural network structures, especially for learning lightweight neural architectures in resource-constrained settings.

## 1 Introduction

Deep neural networks (DNNs) have achieved remarkable empirical successes recently. However, efficient and automatic optimization of model architectures remains to be a key challenge. Compared with parameter optimization which has been well addressed by gradient-based methods (a.k.a. back-propagation), optimizing model structures involves significantly more challenging discrete optimization with large search spaces and high evaluation cost. Although there have been rapid progresses recently, designing the best architectures still requires a lot of expert knowledge and trial-and-errors for most practical tasks.

This work targets extending the power of gradient descent to the domain of model structure optimization of neural networks. In particular, we consider the problem of progressively growing a neural network by "splitting" existing neurons into several "off-springs", and develop a simple and practical approach for deciding the best subset of neurons to split and how to split them, adaptively based on the existing model structure. We derive the optimal splitting strategies by considering the *steepest descent* of the loss when the off-springs are infinitesimally close to the original neurons, yielding a *splitting steepest descent* that monotonically decrease the loss in the space of model structures.

Our main method, shown in Algorithm 1, alternates between a standard *parametric descent phase* which updates the parameters to minimize the loss with a fixed model structure, and a *splitting phase* which updates the model structures by splitting neurons. The splitting phase is triggered when no further improvement can be made by only updating parameters, and allow us to escape the parametric local optima by augmenting the neural network in a locally optimal fashion. Theoretically, these two phases can be viewed as performing functional steepest descent on an $\infty$-Wasserstein metric space, in which the splitting phase is a *second-order descent* for escaping saddle points in the functional space, while the parametric gradient descent corresponds to a *first-order descent*. Empirically, our algorithm is simple and practical, and provides a promising tool for many challenging problems, including progressive training of interpretable neural networks, learning lightweight and energy-efficient neural architectures for resource-constrained settings, and transfer learning, etc.

---

[*]Equal contribution

**Related Works**  The idea of progressively growing neural networks by node splitting is not new, but previous works are mostly based on heuristic or purely random splitting strategies (e.g., Wynne-Jones, 1992; Chen et al., 2016). A different approach for progressive training is the Frank-Wolfe or gradient boosting based strategies (e.g., Schwenk & Bengio, 2000; Bengio et al., 2006; Bach, 2017), which iteratively add new neurons derived from functional conditional gradient, while keeping the previous neurons fixed. However, these methods are not suitable for large scale settings, because adding each neuron requires to solve a difficult non-convex optimization problem, and keeping the previous neurons fixed prevents us from correcting the mistakes made in earlier iterations. A practical alternative of Frank-Wolfe is to simply add new randomly initialized neurons and co-optimize the new and old neurons together. However, random initialization does not allow us to leverage the information of the existing model and takes more time to converge. In contrast, splitting neurons from the existing network allows us to inherent the knowledge from the existing model (see Chen et al. (2016)), and is faster to converge in settings like continual learning, when the previous model is not far away from the optimal solution.

An opposite direction of progressive training is to prune large pre-trained networks (e.g., Han et al., 2016; Li et al., 2017; Liu et al., 2017). In comparison, our splitting method requires no large pre-trained models and can outperform existing pruning methods in terms of learning ultra-small neural architectures, which is of critical importance for resource-constrained settings like mobile devices and Internet of things. More broadly, there has been a series of recent works on neural architecture search, based on various strategies from combinatorial optimization, including reinforcement learning (RL) (e.g., Pham et al., 2018; Cai et al., 2018; Zoph & Le, 2017), evolutionary algorithms (EA) (e.g., Stanley & Miikkulainen, 2002; Real et al., 2018), and continuous relaxation (e.g., Liu et al., 2019a; Xie et al., 2018). However, these general-purpose black-box optimization methods do not leverage the inherent geometric structure of the loss landscape, and are highly computationally expensive due to the need of evaluating the candidate architectures based on inner training loops.

**Background: Steepest Descent and Saddle Points**  Stochastic gradient descent is the driving horse for solving large scale optimization in machine learning and deep learning. Gradient descent can be viewed as a steepest descent procedure that iteratively improves the solution by following the direction that maximally decreases the loss function within a small neighborhood of the previous solution. Specifically, for minimizing a loss function $L(\theta)$, each iteration of steepest descent updates the parameter via $\theta \leftarrow \theta + \epsilon\delta$, where $\epsilon$ is a small step size and $\delta$ is an update direction chosen to maximally decrease the loss $L(\theta + \epsilon\delta)$ of the updated parameter under a norm constraint $\|\delta\| \leq 1$, where $\|\cdot\|$ denotes the Euclidean norm. When $\nabla L(\theta) \neq 0$ and $\epsilon$ is infinitesimal, the optimal descent direction $\delta$ equals the negative gradient direction, that is, $\delta = -\nabla L(\theta)/\|\nabla L(\theta)\|$, yielding a descent of $L(\theta + \epsilon\delta) - L(\theta) \approx -\epsilon\|\nabla L(\theta)\|$. At a critical point with a zero gradient ($\nabla L(\theta) = 0$), the steepest descent direction depends on the spectrum of the Hessian matrix $\nabla^2 L(\theta)$. Denote by $\lambda_{min}$ the minimum eigenvalue of $\nabla^2 L(\theta)$ and $v_{min}$ its associated eigenvector. When $\lambda_{min} > 0$, the point $\theta$ is a stable local minimum and no further improvement can be made in the infinitesimal neighborhood. When $\lambda_{min} < 0$, the point $\theta$ is a saddle point or local maximum, and the steepest descent direction equals the eigenvector $\pm v_{min}$, which yields an $\epsilon^2\lambda_{min}/2$ decrease on the loss.[2] In practice, it has been shown that there is no need to explicitly calculate the negative eigenvalue direction, because saddle points and local maxima are unstable and can be escaped by using gradient descent with random initialization or stochastic noise (e.g., Lee et al., 2016; Jin et al., 2017).

## 2  Splitting Neurons Using Steepest Descent

We introduce our main method in this section. We first illustrate the idea with the simple case of splitting a single neuron in Section 2.1, and then consider the more general case of simultaneously splitting multiple neurons in deep networks in Section 2.2, which yields our main progressive training algorithm (Algorithm 1). Section 2.3 draws a theoretical discussion and interpret our procedure as a functional steepest descent of the distribution of the neuron weights under the $\infty$-Wasserstein metric.

## 2.1 Splitting a Single Neuron

Let $\sigma(\theta, x)$ be a neuron inside a neural network that we want to learn from data, where $\theta$ is the parameter of the neuron and $x$ its input variable. Assume the loss of $\theta$ has a general form of

$$L(\theta) := \mathbb{E}_{x \sim \mathcal{D}}[\Phi(\sigma(\theta, x))], \tag{1}$$

where $\mathcal{D}$ is a data distribution, and $\Phi$ is a map determined by the overall loss function. The parameters of the other parts of the network are assumed to be fixed or optimized using standard procedures and are omitted for notation convenience.

Standard gradient descent can only yield parametric updates of $\theta$. We introduce a generalized steepest descent procedure that allows us to incrementally grow the neural network by gradually introducing new neurons, achieved by "splitting" the existing neurons into multiple copies in a (locally) optimal fashion derived using ideas from steepest descent idea.

In particular, we split $\theta$ into $m$ off-springs $\boldsymbol{\theta} := \{\theta_i\}_{i=1}^m$, and replace the neuron $\sigma(\theta, x)$ with a weighted sum of the off-spring neurons $\sum_{i=1}^m w_i \sigma(\theta_i, x)$, where $\boldsymbol{w} := \{w_i\}_{i=1}^m$ is a set of positive weights assigned on the off-springs, and satisfies $\sum_{i=1}^m w_i = 1$, $w_i > 0$. This yields an augmented loss function on $\boldsymbol{\theta}$ and $\boldsymbol{w}$:

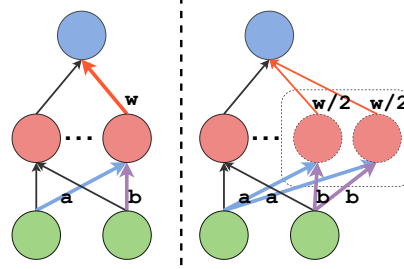

$$\mathcal{L}(\boldsymbol{\theta}, \boldsymbol{w}) := \mathbb{E}_{x \sim \mathcal{D}} \left[ \Phi \left( \sum_{i=1}^m w_i \sigma(\theta_i, x) \right) \right]. \tag{2}$$

A key property of this construction is that it introduces a smooth change on the loss function when the off-springs $\{\theta_i\}_{i=1}^m$ are close to the original parameter $\theta$: when $\theta_i = \theta$, $\forall i = 1, \ldots, m$, the augmented network and loss are equivalent to the original ones, that is, $\mathcal{L}(\theta \mathbf{1}_m, \boldsymbol{w}) = L(\theta)$, where $\mathbf{1}_m$ denotes the $m \times 1$ vector consisting of all ones; when all the $\{\theta_i\}$ are within an infinitesimal neighborhood of $\theta$, it yields an infinitesimal change on the loss, with which a steepest descent can be derive.

Formally, consider the set of splitting schemes $(m, \boldsymbol{\theta}, \boldsymbol{w})$ whose off-springs are $\epsilon$-close to the original neuron:

$$\{(m, \boldsymbol{\theta}, \boldsymbol{w}) : m \in \mathbb{N}_+, \; \|\theta_i - \theta\| \leq \epsilon, \; \sum_{i=1}^m w_i = 1, \; w_i > 0, \; \forall i = 1, \ldots, m\}.$$

We want to decide the optimal $(m, \boldsymbol{\theta}, \boldsymbol{w})$ to maximize the decrease of loss $\mathcal{L}(\boldsymbol{\theta}, \boldsymbol{w}) - L(\theta)$, when the step size $\epsilon$ is infinitesimal. Although this appears to be an infinite dimensional optimization because $m$ is allowed to be arbitrarily large, we show that the optimal choice is achieved with either $m = 1$ (no splitting) or $m = 2$ (splitting into two off-springs), with uniform weights $w_i = 1/m$. Whether a neuron should be split ($m = 1$ or $2$) and the optimal values of the off-springs $\{\theta_i\}$ are decided by the minimum eigenvalue and eigenvector of a *splitting matrix*, which plays a role similar to Hessian matrix for deciding saddle points.

**Definition 2.1 (Splitting Matrix).** *For $L(\theta)$ in (1), its **splitting matrix** $S(\theta)$ is defined as*

$$S(\theta) = \mathbb{E}_{x \sim \mathcal{D}}[\Phi'(\sigma(\theta, x)) \nabla^2_{\theta\theta} \sigma(\theta, x)]. \tag{3}$$

*We call the minimum eigenvalue $\lambda_{min}(S(\theta))$ of $S(\theta)$ the **splitting index** of $\theta$, and the eigenvector $v_{min}(S(\theta))$ related to $\lambda_{min}(S(\theta))$ the **splitting gradient** of $\theta$.*

The splitting matrix $S(\theta)$ is a $\mathbb{R}^{d \times d}$ symmetric "semi-Hessian" matrix that involves the first derivative $\Phi'(\cdot)$, and the second derivative of $\sigma(\theta, x)$. It is useful to compare it with the typical gradient and Hessian matrix of $L(\theta)$:

$$\nabla_\theta L(\theta) = \mathbb{E}_{x \sim \mathcal{D}}[\Phi'(\sigma(\theta, x)) \nabla_\theta \sigma(\theta, x)], \qquad \nabla^2_{\theta\theta} L(\theta) = S(\theta) + \underbrace{\mathbb{E}[\Phi''(\sigma(\theta, x)) \nabla_\theta \sigma(\theta, x)^{\otimes 2}]}_{T(\theta)},$$

where $v^{\otimes 2} := vv^\top$ is the outer product. The splitting matrix $S(\theta)$ differs from the gradient $\nabla_\theta L(\theta)$ in replacing $\nabla_\theta \sigma(\theta, x)$ with the second-order derivative $\nabla^2_{\theta\theta} \sigma(\theta, x)$, and differs from the Hessian

matrix $\nabla^2_{\theta\theta}L(\theta)$ in missing an extra term $T(\theta)$. We should point out that $S(\theta)$ is the "easier part" of the Hessian matrix, because the second-order derivative $\nabla^2_{\theta\theta}\sigma(\theta, x)$ of the individual neuron $\sigma$ is much simpler than the second-order derivative $\Phi''(\cdot)$ of "everything else", which appears in the extra term $T(\theta)$. In addition, as we show in Section 2.2, $S(\theta)$ is block diagonal in terms of multiple neurons, which is crucial for enabling practical computational algorithm.

It is useful to decompose each $\theta_i$ into $\theta_i = \theta + \epsilon(\mu + \delta_i)$, where $\mu$ is an average displacement vector shared by all copies, and $\delta_i$ is the splitting vector associated with $\theta_i$, and satisfies $\sum_i w_i\delta_i = 0$ (which implies $\sum_i w_i\theta_i = \theta + \epsilon\mu$). It turns out that the change of loss $\mathcal{L}(\boldsymbol{\theta}, \boldsymbol{w}) - L(\theta)$ naturally decomposes into two terms that reflect the effects of the average displacement and splitting, respectively.

**Theorem 2.2.** *Assume $\theta_i = \theta + \epsilon(\mu + \delta_i)$ with $\sum_i w_i\delta_i = 0$ and $\sum_i w_i = 1$. For $L(\theta)$ and $\mathcal{L}(\boldsymbol{\theta}, \boldsymbol{w})$ in (1) and (2), assume $\mathcal{L}(\boldsymbol{\theta}, \boldsymbol{w})$ has bounded third order derivatives w.r.t. $\boldsymbol{\theta}$. We have*

$$\mathcal{L}(\boldsymbol{\theta}, \boldsymbol{w}) - L(\theta) = \underbrace{\epsilon\nabla L(\theta)^\top \mu + \frac{\epsilon^2}{2}\mu^\top\nabla^2 L(\theta)\mu}_{I(\mu; \theta) = L(\theta + \epsilon\mu) - L(\theta) + \mathcal{O}(\epsilon^3)} + \underbrace{\frac{\epsilon^2}{2}\sum_{i=1}^m w_i\delta_i^\top S(\theta)\delta_i}_{II(\boldsymbol{\delta}, \boldsymbol{w}; \theta)} + \mathcal{O}(\epsilon^3), \quad (4)$$

*where the change of loss is decomposed into two terms: the first term $I(\mu; \theta)$ is the effect of the average displacement $\mu$, and it is equivalent to applying the standard parametric update $\theta \leftarrow \theta + \epsilon\mu$ on $L(\theta)$. The second term $II(\boldsymbol{\delta}, \boldsymbol{w}; \theta)$ is the change of the loss caused by the splitting vectors $\boldsymbol{\delta} := \{\delta_i\}$. It depends on $L(\theta)$ only through the splitting matrix $S(\theta)$.*

Therefore, the optimal average displacement $\mu$ should be decided by standard parametric steepest (gradient) descent, which yields a typical $\mathcal{O}(\epsilon)$ decrease of loss at non-stationary points. In comparison, the splitting term $II(\boldsymbol{\delta}, \boldsymbol{w}; \theta)$ is always $\mathcal{O}(\epsilon^2)$, which is much smaller. Given that introducing new neurons increases model size, splitting should not be preferred unless it is impossible to achieve an $\mathcal{O}(\epsilon^2)$ gain with pure parametric updates that do not increase the model size. Therefore, it is motivated to introduce splitting only at stable local minima, when the optimal $\mu$ equals zero and no further improvement is possible with (infinitesimal) regular parametric descent on $L(\theta)$. In this case, we only need to minimize the splitting term $II(\boldsymbol{\delta}, \boldsymbol{w}; \theta)$ to decide the optimal splitting strategy, which is shown in the following theorem.

**Theorem 2.3.** *a) If the splitting matrix is positive definite, that is, $\lambda_{min}(S(\theta)) > 0$, we have $II(\boldsymbol{\delta}, \boldsymbol{w}; \theta) > 0$ for any $\boldsymbol{w} > 0$ and $\boldsymbol{\delta} \neq 0$, and hence no infinitesimal splitting can decrease the loss. We call that $\theta$ is splitting stable in this case.*

*b) If $\lambda_{min}(S(\theta)) < 0$, an optimal splitting strategy that minimizes $II(\boldsymbol{\delta}, \boldsymbol{w}; \theta)$ subject to $\|\delta_i\| \leq 1$ is*

$$m = 2, \qquad w_1 = w_2 = 1/2, \qquad and \qquad \delta_1 = v_{min}(S(\theta)), \qquad \delta_2 = -v_{min}(S(\theta)),$$

*where $v_{min}(S(\theta))$, called the splitting gradient, is the eigenvector related to $\lambda_{min}(S(\theta))$. Here we split the neuron into two copies of equal weights, and update each copy with the splitting gradient. The change of loss obtained in this case is $II(\{\delta_1, -\delta_1\}, \{1/2, 1/2\}; \theta) = -\epsilon^2\lambda_{min}(S(\theta))/2 < 0$.*

**Remark** The splitting stability ($S(\theta) \succ 0$) does not necessarily ensure the standard parametric stability of $L(\theta)$ (i.e., $\nabla^2 L(\theta) = S(\theta) + T(\theta) \succ 0$), except when $\Phi(\cdot)$ is convex which ensures $T(\theta) \succeq 0$ (see Definition 2.1). If both $S(\theta) \succ 0$ and $\nabla^2 L(\theta) \succ 0$ hold, the loss can not be improved by any local update or splitting, no matter how many off-springs are allowed. Since stochastic gradient descent guarantees to escape unstable stationary points (Lee et al., 2016; Jin et al., 2017), we only need to calculate $S(\theta)$ to decide the splitting stability in practice.

## 2.2 Splitting Deep Neural Networks

In practice, we need to split multiple neurons simultaneously, which may be of different types, or locate in different layers of a deep neural network. The key questions are if the optimal splitting strategies of different neurons influence each other in some way, and how to compare the gain of splitting different neurons and select the best subset of neurons to split under a budget constraint.

It turns out the answers are simple. We show that the change of loss caused by splitting a set of neurons is simply the sum of the splitting terms $II(\boldsymbol{\delta}, \boldsymbol{w}; \theta)$ of the individual neurons. Therefore, we

---

**Algorithm 1** Splitting Steepest Descent for Optimizing Neural Architectures

---

**Initialize** a neural network with a set of neurons $\theta^{[1:n]} = \{\theta^{[\ell]}\}_{\ell=1}^{n}$ that can be split, whose loss satisfies (5). Decide a maximum number $m_*$ of neurons to split at each iteration, and a threshold $\lambda_* \leq 0$ of the splitting index. A stepsize $\epsilon$.

**1. Update the parameters** using standard optimizers (e.g., stochastic gradient descent) until no further improvement can be made by only updating parameters.

**2. Calculate the splitting matrices** $\{S^{[\ell]}\}$ of the neurons following (7), as well as their minimum eigenvalues $\{\lambda_{min}^{[\ell]}\}$ and the associated eigenvectors $\{v_{min}^{[\ell]}\}$.

**3. Select the set of neurons to split** by picking the top $m_*$ neurons with the smallest eigenvalues $\{\lambda_{min}^{[\ell]}\}$ and satisfies $\lambda_{min}^{[\ell]} \leq \lambda_*$.

**4. Split each of the selected neurons** into two off-springs with equal weights, and update the neuron network by replacing each selected neuron $\sigma_\ell(\theta^{[\ell]}, \cdot)$ with

$$\frac{1}{2}(\sigma_\ell(\theta_1^{[\ell]}, \cdot) + \sigma_\ell(\theta_2^{[\ell]}, \cdot)), \qquad \text{where} \qquad \theta_1^{[\ell]} \leftarrow \theta^{[\ell]} + \epsilon v_{min}^{[\ell]}, \qquad \theta_2^{[\ell]} \leftarrow \theta^{[\ell]} - \epsilon v_{min}^{[\ell]}.$$

Update the list of neurons. Go back to Step 1 or stop when a stopping criterion is met.

---

can calculate the splitting matrix of each neuron independently without considering the other neurons, and compare the "splitting desirability" of the different neurons by their minimum eigenvalues (splitting indexes). This motivates our main algorithm (Algorithm 1), in which we progressively split the neurons with the most negative splitting indexes following their own splitting gradients. Since the neurons can be in different layers and of different types, this provides an adaptive way to grow neural network structures to fit best with data.

To set up the notation, let $\theta^{[1:n]} = \{\theta^{[1]}, \ldots \theta^{[n]}\}$ be the parameters of a set of neurons (or any duplicable sub-structures) in a large neural network, where $\theta^{[\ell]}$ is the parameter of the $\ell$-th neuron. Assume we split $\theta^{[\ell]}$ into $m_\ell$ copies $\boldsymbol{\theta}^{[\ell]} := \{\theta_i^{[\ell]}\}_{i=1}^{m_\ell}$, with weights $\boldsymbol{w}^{[\ell]} = \{w_i^{[\ell]}\}_{i=1}^{m_\ell}$ satisfying $\sum_{i=1}^{m_\ell} w_i^{[\ell]} = 1$ and $w_i^{[\ell]} \geq 0$, $\forall i = 1, \ldots, m_\ell$. Denote by $L(\theta^{[1:n]})$ and $\mathcal{L}(\boldsymbol{\theta}^{[1:n]}, \boldsymbol{w}^{[1:n]})$ the loss function of the original and augmented networks, respectively. It is hard to specify the actual expression of the loss functions in general cases, but it is sufficient to know that $L(\theta^{[1:n]})$ depends on each $\theta^{[\ell]}$ only through the output of its related neuron,

$$L(\theta^{[1:n]}) = \mathbb{E}_{x \sim \mathcal{D}} \left[ \Phi_\ell \left( \sigma_\ell \left( \theta^{[\ell]}, \ h^{[\ell]} \right); \ \theta^{[\neg \ell]} \right) \right], \qquad h^{[\ell]} = g_\ell(x; \ \theta^{[\neg \ell]}), \qquad (5)$$

where $\sigma_\ell$ denotes the activation function of neuron $\ell$, and $g_\ell$ and $\Phi_\ell$ denote the parts of the loss that connect to the input and output of neuron $\ell$, respectively, both of which depend on the other parameters $\theta^{[\neg \ell]}$ in some complex way. Similarly, the augmented loss $\mathcal{L}(\boldsymbol{\theta}^{[1:n]}, \boldsymbol{w}^{[1:n]})$ satisfies

$$\mathcal{L}(\boldsymbol{\theta}^{[1:n]}, \boldsymbol{w}^{[1:n]}) = \mathbb{E}_{x \sim \mathcal{D}} \left[ \boldsymbol{\Phi}_\ell \left( \sum_{i=1}^{m_\ell} w_i \sigma_\ell \left( \theta_i^{[\ell]}, \ \boldsymbol{h}^{[\ell]} \right); \ \boldsymbol{\theta}^{[\neg \ell]}, \boldsymbol{w}^{[\neg \ell]} \right) \right], \qquad (6)$$

where $\boldsymbol{h}^{[\ell]} = \boldsymbol{g}_\ell(x; \ \boldsymbol{\theta}^{[\neg \ell]}, \boldsymbol{w}^{[\neg \ell]})$, and $\boldsymbol{g}_\ell$, $\boldsymbol{\Phi}_\ell$ are the augmented variants of $g_\ell$, $\Phi_\ell$, respectively.

Interestingly, although each equation in (5) and (6) only provides a partial specification of the loss function of deep neural nets, they together are sufficient to establish the following key extension of Theorem 2.2 to the case of multiple neurons.

**Theorem 2.4.** *Under the setting above, assume* $\theta_i^{[\ell]} = \theta^{[\ell]} + \epsilon(\mu^{[\ell]} + \delta_i^{[\ell]})$ *for* $\forall \ell \in [1:n]$, *where* $\mu^{[\ell]}$ *denotes the average displacement vector on* $\theta^{[\ell]}$, *and* $\delta_i^{[\ell]}$ *is the* $i$-*th splitting vector of* $\theta^{[\ell]}$, *with* $\sum_{i=1}^{m_\ell} w_i \delta_i^{[\ell]} = 0$. *Assume* $\mathcal{L}(\boldsymbol{\theta}^{[1:n]}, \boldsymbol{w}^{[1:n]})$ *has bounded third order derivatives w.r.t.* $\boldsymbol{\theta}^{[1:n]}$. *We have*

$$\mathcal{L}(\boldsymbol{\theta}^{[1:n]}, \boldsymbol{w}^{[1:n]}) = L(\theta^{[1:n]} + \epsilon \mu^{[1:n]}) + \sum_{\ell=1}^{n} \underbrace{\frac{\epsilon^2}{2} \sum_{i=1}^{m_\ell} w_i^{[\ell]} {\delta_i^{[\ell]}}^\top S^{[\ell]}(\theta^{[1:n]}) \delta_i^{[\ell]}}_{II_\ell(\boldsymbol{\delta}^{[\ell]}, \boldsymbol{w}^{[\ell]}; \ \theta^{[1:n]})} + \mathcal{O}(\epsilon^3),$$

*where the effect of average displacement is again equivalent to that of the corresponding parametric update $\theta^{[1:n]} \leftarrow \theta^{[1:n]} + \epsilon\mu^{[1:n]}$; the splitting effect equals the sum of the individual splitting terms $\Pi_\ell(\boldsymbol{\delta}^{[\ell]}, \boldsymbol{w}^{[\ell]}; \theta^{[1:n]})$, which depends on the splitting matrix $S^{[\ell]}(\theta^{[1:n]})$ of neuron $\ell$,*

$$S^{[\ell]}(\theta^{[1:n]}) = \mathbb{E}_{x \sim \mathcal{D}} \left[ \nabla_{\sigma_\ell} \Phi_\ell \left( \sigma_\ell \left( \theta^{[\ell]}, \; h^{[\ell]} \right); \; \theta^{[\neg\ell]} \right) \nabla^2_{\theta\theta} \sigma_\ell \left( \theta^{[\ell]}, \; h^{[\ell]} \right) \right]. \tag{7}$$

The important implication of Theorem 2.4 is that there is *no crossing term* in the splitting matrix, unlike the standard Hessian matrix. Therefore, the splitting effect of an individual neuron only depends on its own splitting matrix and can be evaluated individually; the splitting effects of different neurons can be compared using their splitting indexes, allowing us to decide the best subset of neurons to split when a maximum number constraint is imposed. As shown in Algorithm 1, we decide a maximum number $m_*$ of neurons to split at each iteration, and a threshold $\lambda_* \leq 0$ of splitting index, and split the neurons whose splitting indexes are ranked in top $m_*$ and smaller than $\lambda_*$.

**Computational Efficiency**   The computational cost of exactly evaluating all the splitting indexes and gradients on a data instance is $\mathcal{O}(nd^3)$, where $n$ is the number of neurons and $d$ is the number of the parameters of each neuron. Note that this is much better than evaluating the Hessian matrix, which costs $\mathcal{O}(N^3)$, where $N$ is the total number of parameters (e.g., $N \geq nd$). In practice, $d$ is not excessively large or can be controlled by identifying a subset of important neurons to split. Further computational speedup can be obtained by using efficient gradient-based large scale eigen-computation methods, which we investigate in future work.

### 2.3  Splitting as $\infty$-Wasserstein Steepest Descent

We present a functional aspect of our approach, in which we frame the co-optimization of the neural parameters and structures into a functional optimization in the space of distributions of the neuron weights, and show that our splitting strategy can be viewed as a second-order descent for escaping saddle points in the $\infty$-Wasserstein space of distributions, while the standard parametric gradient descent corresponds to a first-order descent in the same space.

We illustrate our theory using the single neuron case in Section 2.1. Consider the augmented loss $\mathcal{L}(\boldsymbol{\theta}, \boldsymbol{w})$ in (2). Because the off-springs of the neuron are exchangeable, we can equivalently represent $\mathcal{L}(\boldsymbol{\theta}, \boldsymbol{w})$ as a functional of the empirical measure of the off-springs,

$$\mathcal{L}[\rho] = \mathbb{E}_{x \sim \mathcal{D}} \left[ \Phi \left( \mathbb{E}_{\theta \sim \rho}[\sigma(\theta, x)] \right) \right], \qquad\qquad \rho = \sum_{i=1}^{m} w_i \delta_{\theta_i}, \tag{8}$$

where $\delta_{\theta_i}$ denotes the delta measure on $\theta_i$ and $\mathcal{L}[\rho]$ is the functional representation of $\mathcal{L}(\boldsymbol{\theta}, \boldsymbol{w})$. The idea is to optimize $\mathcal{L}[\rho]$ in the space of probability distributions (or measures) using a functional steepest descent. To do so, a notion of distance on the space of distributions need to be decided. We consider the $p$-Wasserstein metric,

$$\mathbb{D}_p(\rho, \rho') = \inf_{\gamma \in \Pi(\rho, \rho')} \left( \mathbb{E}_{(\theta, \theta') \sim \gamma}[\|\theta - \theta'\|^p] \right)^{1/p}, \qquad\qquad \text{for } p > 0, \tag{9}$$

where $\Pi(\rho, \rho')$ denotes the set of probability measures whose first and second marginals are $\rho$ and $\rho'$, respectively, and $\gamma$ can be viewed as describing a transport plan from $\rho$ to $\rho'$. We obtain the $\infty$-Wasserstein metric $\mathbb{D}_\infty(\rho, \rho')$ in the limit when $p \to +\infty$, in which case the $p$-norm reduces to an esssup norm, that is,

$$\mathbb{D}_\infty(\rho, \rho') = \inf_{\gamma \in \Pi(\rho, \rho')} \operatorname*{esssup}_{(\theta, \theta') \sim \gamma} [\|\theta - \theta'\|],$$

where the esssup notation denotes the smallest number $c$ such that the set $\{(\theta, \theta'): \|\theta - \theta'\| > c\}$ has zero probability under $\gamma$. See more discussion in Villani (2008) and Appendix A.2.

The $\infty$-Wasserstein metric yields a natural connection to node splitting. For each $\theta$, the conditional distribution $\gamma(\theta' \mid \theta)$ represents the distribution of points $\theta'$ transported from $\theta$, which can be viewed as the off-springs of $\theta$ in the context of node splitting. If $\mathbb{D}_\infty(\rho, \rho') \leq \epsilon$, it means that $\rho'$ can be obtained from splitting $\theta \sim \rho$ such that all the off-springs are $\epsilon$-close, i.e., $\|\theta' - \theta\| \leq \epsilon$. This is consistent with the augmented neighborhood introduced in Section 2.1, except that $\gamma$ here can be an absolutely continuous distribution, representing a continuously infinite number of off-springs; but this

yields no practical difference because any distribution $\gamma$ can be approximated arbitrarily close using a countable number of particles. Note that $p$-Wasserstein metrics with finite $p$ are not suitable for our purpose because $\mathbb{D}_p(\rho, \rho') \leq \epsilon$ with $p < \infty$ does not ensure $\|\theta' - \theta\| \leq \epsilon$ for all $\theta \sim \rho$ and $\theta' \sim \rho'$.

Similar to the steepest descent on the Euclidean space, the $\infty$-Wasserstein steepest descent on $\mathcal{L}[\rho]$ should iteratively find new points that maximize the decrease of loss in an $\epsilon$-ball of the current points. Define

$$\rho^* = \arg\min_{\rho'}\{\mathcal{L}[\rho'] - \mathcal{L}[\rho]\colon \quad \mathbb{D}_\infty(\rho, \rho') \leq \epsilon\}, \qquad \Delta^*(\rho, \epsilon) = \mathcal{L}[\rho^*] - \mathcal{L}[\rho].$$

We are ready to show the connection of Algorithm 1 to the $\infty$-Wasserstein steepest descent.

**Theorem 2.5.** *Consider the $\mathcal{L}(\boldsymbol{\theta}, \boldsymbol{w})$ and $\mathcal{L}[\rho]$ in (2) and (8), connected with $\rho = \sum_i w_i \delta_{\theta_i}$. Define $G_\rho(\theta) = \mathbb{E}_{x \sim \mathcal{D}}[\Phi'(f_\rho(x))\nabla_\theta \sigma(\theta, x)]$ and $S_\rho(\theta) = \mathbb{E}_{x \sim \mathcal{D}}[\Phi'(f_\rho(x))\nabla^2_{\theta\theta}\sigma(\theta, x)]$ with $f_\rho(x) = \mathbb{E}_{\theta \sim \rho}[\sigma(\theta, x)]$, which are related to the gradient and splitting matrices of $\mathcal{L}(\boldsymbol{\theta}, \boldsymbol{w})$, respectively. Assume $\mathcal{L}(\boldsymbol{\theta}, \boldsymbol{w})$ has bounded third order derivatives w.r.t. $\boldsymbol{\theta}$.*

*a) If $\mathcal{L}(\boldsymbol{\theta}, \boldsymbol{w})$ is on a non-stationary point w.r.t. $\boldsymbol{\theta}$, then the steepest descent of $\mathcal{L}[\rho]$ is achieved by moving all the particles of $\rho$ with gradient descent on $\mathcal{L}(\boldsymbol{\theta}, \boldsymbol{w})$, that is,*

$$\mathcal{L}[(I - \epsilon G_\rho)\sharp\rho] - \mathcal{L}[\rho] = \Delta^*(\rho, \epsilon) + \mathcal{O}(\epsilon^2) = -\epsilon\mathbb{E}_{\theta \sim \rho}[\|G_\rho(\theta)\|] + \mathcal{O}(\epsilon^2),$$

*where $(I - \epsilon G_\rho)\sharp\rho$ denotes the distribution of $\theta' = \theta - \epsilon G_\rho(\theta)/\|G_\rho(\theta)\|$ when $\theta \sim \rho$.*

*b) If $\mathcal{L}(\boldsymbol{\theta}, \boldsymbol{w})$ reaches a stable local optima w.r.t. $\boldsymbol{\theta}$, the steepest descent on $\mathcal{L}[\rho]$ is splitting each neuron with $\lambda_{min}(S_\rho(\theta)) < 0$ into two copies of equal weights following their minimum eigenvectors, while keeping the remaining neurons to be unchanged. Precisely, denote by $(I \pm \epsilon v_{min}(S_\rho(\theta))_+)\sharp\rho$ the distribution obtained in this way, we have*

$$\mathcal{L}[(I \pm \epsilon v_{min}(S_\rho(\theta))_+)\sharp\rho] - \mathcal{L}[\rho] = \Delta^*(\rho, \epsilon) + \mathcal{O}(\epsilon^3),$$

*where we have $\Delta^*(\rho, \epsilon) = \epsilon^2\mathbb{E}_{\theta \sim \rho}[\min(\lambda_{min}(S_\rho(\theta)), 0)]/2$.*

**Remark** There has been a line of theoretical works on analyzing gradient-based learning of neural networks via 2-Wasserstein gradient flow by considering the *mean field limit* when the number of neurons $m$ goes to infinite ($m \to \infty$) (e.g. Mei et al., 2018; Chizat & Bach, 2018). These analysis focus on the first-order descent on the 2-Wasserstein space as a theoretical tool for understanding the behavior of gradient descent on overparameterized neural networks. Our framework is significant different, since we mainly consider the second-order descent on the $\infty$-Wasserstein space, and the case of finite number of neurons $m$ in order to derive practical algorithms.

## 3 Experiments

We test our method on both toy and realistic tasks, including learning interpretable neural networks, architecture search for image classification and energy-efficient keyword spotting. Due to limited space, many of the detailed settings are shown in Appendix, in which we also include additional results on distribution approximation (Appendix C.1), transfer learning (Appendix C.2).

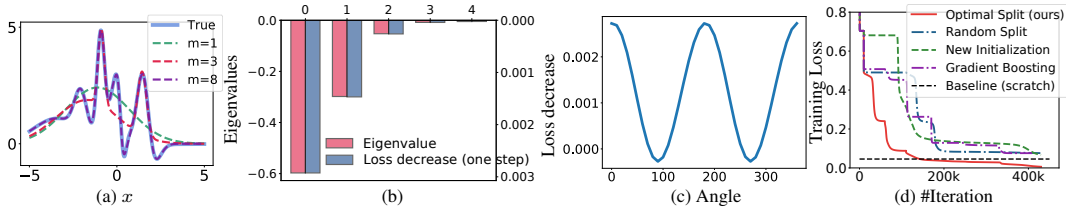

Figure 1: Results on a one-dimensional RBF network. (a) The true and estimated functions. (b) The eigenvalue vs. loss decrease. (c) The loss decrease vs. the angle of the splitting direction with the minimum eigenvector. (d) The training loss vs. the iteration (of gradient descent); the splittings happen at the cliff points.

**Toy RBF Neural Networks**   We apply our method to learn a one-dimensional RBF neural network shown in Figure 1a. See Appendix B.1 for details of the setting. We start with a small neural network with $m = 1$ neuron and gradually increase the model size by splitting neurons. Figure 1a shows that we almost recover the true function as we split up to $m = 8$ neurons. Figure 1b shows the top five eigenvalues and the decrease of loss when we split $m = 7$ neurons to $m = 8$ neurons; we can see that the eigenvalue and loss decrease correlate linearly, confirming our results in Theorem 2.4. Figure 1c shows the decrease of the loss when we split the top one neuron following the direction with different angles from the minimum eigenvector at $m = 7$. We can see that the decrease of the loss is maximized when the splitting direction aligns with the eigenvector, consistent with our theory. In Figure 1d, we compare with different baselines of progressive training, including `Random Split`, splitting a randomly chosen neuron with a random direction; `New Initialization`, adding a new neuron with randomly initialized weights and co-optimization it with previous neurons; `Gradient Boosting`, adding new neurons with Frank-Wolfe algorithm while fixing the previous neurons; `Baseline (scratch)`, training a network of size $m = 8$ from scratch. Figure 1d shows our method yields the best result.

**Learning Interpretable Neural Networks**   To visualize the dynamics of the splitting process, we apply our method to incrementally train an interpretable neural network designed by Li et al. (2018), which contains a "prototype layer" whose weights are enforced to be similar to realistic images to encourage interpretablity. See Appendix B.2 and Li et al. (2018) for more detailed settings. We apply our method to split the prototype layer starting from a single neuron on MNIST, and show in Figure 2 the evolutionary tree of the neurons in our splitting process. We can see that the blurry (and hence less interpretable) prototypes tend to be selected and split into two off-springs that are similar yet more interpretable. Figure 2 (b) shows the decrease of loss when we split each of the five neurons at the 5-th step (with the decrease of loss measured at the local optima reached dafter splitting); we find that the eigenvalue correlates well with the decrease of loss and the interpretablity of the neurons. The complete evolutionary tree and quantitative comparison with baselines are shown in Appendix B.2.

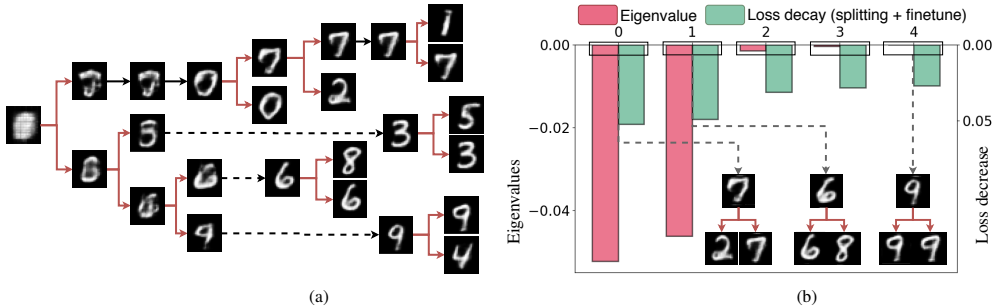

(a)                                                          (b)

Figure 2: Progressive learning of the interpretable prototype network in Li et al. (2018) on MNIST. (a) The evolutionary tree of our splitting process, in which the least interpretable, most ambiguous prototypes tend to be split first. (b) The eigenvalue and resulting loss decay when splitting the different neurons at the 5-th step.

**Lightweight Neural Architectures for Image Classification**   We investigate the effectiveness of our methods in learning small and efficient network structures for image classification. We experiment with two popular deep neural architectures, MobileNet (Howard et al., 2017) and VGG19 (Simonyan & Zisserman, 2015). In both cases, we start with a relatively small network and gradually grow the network by splitting the convolution filters following Algorithm 1. See Appendix B.3 for more details of the setting. Because there is no other off-the-shelf progressive growing algorithm that can adaptively decide the neural architectures like our method, we compare with pruning methods, which follow the opposite direction of gradually removing neurons starting from a large pre-trained network. We test two state-of-the-art pruning methods, including batch-normalization-based pruning (Bn-prune) (Liu et al., 2017) and L1-based pruning (L1-prune) (Li et al., 2017). As shown in Figure 3a-b, our splitting method yields higher accuracy with similar model sizes. This is surprising and significant, because the pruning methods leverage the knowledge from a large pre-train model, while our method does not.

To further test the effect of architecture learning in both splitting and pruning methods, we test another setting in which we discard the weights of the neurons and retain the whole network starting from a random initialization, under the structure obtained from splitting or pruning at each iteration.

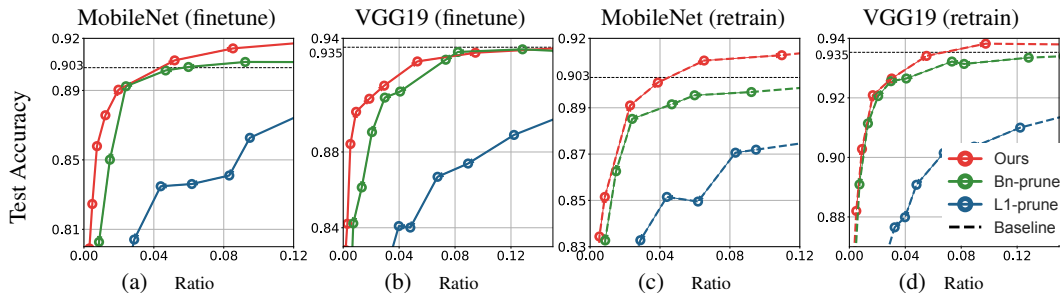

Figure 3: Results on CIFAR-10. (a)-(b) Results of Algorithm 1 and pruning methods (which successively finetune the neurons after pruning). (c)-(d) Results of Algorithm 1 and prunning methods with retrainning, in which we retrain all the weights starting from random initialization after each splitting or pruning step. The x-axis represents the ratio between the number parameters of the learned models and a full size baseline network.

As shown in Figure 3c-d, the results of retraining is comparable with (or better than) the result of successive finetuning in Figure 3a-b, which is consistent with the findings in Liu et al. (2019b). Meanwhile, our splitting method still outperforms both Bn-prune and L1-prune.

**Resource-Efficient Keyword Spotting on Edge Devices** Keyword spotting systems aim to detect a particular keyword from a continuous stream of audio. It is typically deployed on energy-constrained edge devices and requires real-time response and high accuracy for good user experience. This casts a key challenge of constructing efficient and lightweight neural architectures. We apply our method to solve this problem, by splitting a small model (a compact version of DS-CNN) obtained from Zhang et al. (2017). See Appendix B.4 for detailed settings.

Table 1 shows the results on the Google speech commands benchmark dataset (Warden, 2018), in which our method achieves significantly higher accuracy than the best model (DS-CNN) found by Zhang et al. (2017), while having 31% less parameters and Flops. Figure 4 shows further comparison with Bn-prune (Liu et al., 2017), which is again inferior to our method.

| Method | Acc | Params (K) | Ops (M) |
|---|---|---|---|
| DNN | 86.94 | 495.7 | 1.0 |
| CNN | 92.64 | 476.7 | 25.3 |
| BasicLSTM | 93.62 | 492.6 | 47.9 |
| LSTM | 94.11 | 495.8 | 48.4 |
| GRU | 94.72 | 498.0 | 48.4 |
| CRNN | 94.21 | 485.0 | 19.3 |
| DS-CNN | 94.85 | 413.7 | 56.9 |
| Ours | **95.36** | **282.6** | **39.2** |

Table 1: Results on keyword spotting. All results are averaged over 5 rounds.

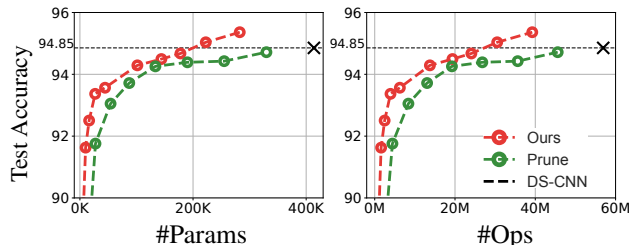

Figure 4: Comparison of accuracy vs. model size (#Params) and number of flops (#Ops) on keyword spotting.

## 4 Conclusion

We present a simple approach for progressively training neural networks via neuron splitting. Our approach highlights a novel view of neural structure optimization as continuous functional optimization, and yields a practical procedure with broad applications. For future work, we will further investigate fast gradient descent based approximation of large scale eigen-computation and more theoretical analysis, extensions and applications of our approach.

## Acknowledgement

This work is supported in part by NSF CRII 1830161 and NSF CAREER 1846421. We would like to acknowledge Google Cloud and Amazon Web Services (AWS) for their support.

## Footnotes

[2]The property of the case when $\lambda_{min} = 0$ depends on higher order information.

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
