[Supplementary Material · supply_nips_final.pdf]

# A  Proofs

## A.1  Proofs of Splitting Taylor Expansion

***Proof of Theorem 2.2.***  Taking the gradient of $L(\theta)$ in (1) gives

$$\nabla_\theta L(\theta) = \mathbb{E}[\Phi'(\sigma(\theta, x))\nabla_\theta \sigma(\theta, x)],$$

$$\nabla^2_{\theta\theta} L(\theta) = \mathbb{E}[\Phi'(\sigma(\theta, x))\nabla^2_{\theta\theta}\sigma(\theta, x) + \Phi''(\sigma(\theta, x))\nabla_\theta \sigma(\theta, x)^{\otimes 2}],$$

where $\Phi'(\cdot)$ is the derivative of $\Phi(\cdot)$ (which is a univariate function), and $\nabla_\theta \sigma(\theta, x)^{\otimes 2} := \nabla_\theta \sigma(\theta, x)\nabla_\theta \sigma(\theta, x)^\top$.

When $\theta$ is split into $\{w_i, \theta_i\}_{i=1}^m$, the augmented loss function is

$$\mathcal{L}(\boldsymbol{\theta}, \boldsymbol{w}) = \mathbb{E}\left[\Phi\left(\sum_{i=1}^m w_i \sigma(\theta_i, x)\right)\right],$$

where $\boldsymbol{w} = [w_1, \ldots, w_m]$ and $\boldsymbol{\theta} = [\theta_1, \ldots, \theta_m]$. The weights should satisfy $\sum_{i=1}^m w_i = 1$ and $w_i \geq 0$. In this way, we have $\mathcal{L}(\boldsymbol{\theta}, \boldsymbol{w}) = L(\theta)$ when $\boldsymbol{\theta} = [\theta, \ldots, \theta] = \theta \mathbf{1}_m$.

Taking the gradient of $\mathcal{L}(\boldsymbol{\theta}, \boldsymbol{w})$ w.r.t. $\theta_i$ when $\boldsymbol{\theta} = \theta \mathbf{1}_m$, we have

$$\nabla_{\theta_i} \mathcal{L}(\theta \mathbf{1}_m, \boldsymbol{w}) = \mathbb{E}\left[\Phi'(\sigma(\theta, x)) w_i \nabla_\theta \sigma(\theta, x)\right] = w_i \nabla_\theta L(\theta).$$

Taking the second derivative, we get

$$\nabla_{\theta_i, \theta_i} \mathcal{L}(\theta \mathbf{1}_m, \boldsymbol{w}) = \mathbb{E}\left[\Phi'(\sigma(\theta, x)) w_i \nabla^2_{\theta, \theta}\sigma(\theta, x) + \Phi''(\sigma(\theta, x)) w_i^2 \nabla_\theta \sigma(\theta, x)^{\otimes 2}\right]$$
$$:= w_i A(\theta) + w_i^2 B(\theta),$$

where

$$A(\theta) := \mathbb{E}\left[\Phi'(\sigma(\theta, x))\nabla^2_{\theta, \theta}\sigma(\theta, x)\right], \qquad B(\theta) := \mathbb{E}\left[\Phi''(\sigma(\theta, x))\nabla_\theta \sigma(\theta, x)^{\otimes 2}\right].$$

Note that we have $\nabla^2_{\theta\theta} L(\theta) = A(\theta) + B(\theta)$ following this definition.

For $i \neq j$, we have

$$\nabla_{\theta_i \theta_j} \mathcal{L}(\theta \mathbf{1}_m, \boldsymbol{w}) = \mathbb{E}\left[\Phi''(\sigma(\theta, x)) w_i w_j \nabla_\theta \sigma(\theta, x)^{\otimes 2}\right] = w_i w_j B(\theta).$$

For $\boldsymbol{\theta} = [\theta_1, \ldots, \theta_m]$, assume $\theta_i = \theta + \epsilon \delta_i$, and define $\bar{\delta} = \sum_{i=1}^m w_i \delta_i$ to be the average displacement. Therefore, $\boldsymbol{\theta} = \theta \mathbf{1}_m + \epsilon \boldsymbol{\delta}$. Using the Taylor expansion of $\mathcal{L}(\theta \mathbf{1}_m + \epsilon \boldsymbol{\delta}, \boldsymbol{w})$ w.r.t. $\epsilon$ at $\epsilon = 0$, we have

$$\mathcal{L}(\boldsymbol{\theta}, \boldsymbol{w}) - L(\theta) = \mathcal{L}(\theta \mathbf{1}_m + \epsilon \boldsymbol{\delta}, \boldsymbol{w}) - L(\theta)$$

$$= \epsilon \sum_{i=1}^m \nabla_{\theta_i} \mathcal{L}(\theta \mathbf{1}_m, \boldsymbol{w})^\top \delta_i + \frac{\epsilon^2}{2}\sum_{ij=1}^m \delta_i^\top (\nabla^2_{\theta_i, \theta_j} \mathcal{L}(\theta \mathbf{1}_m, \boldsymbol{w}))\delta_i + \mathcal{O}(\epsilon^3)$$

$$= \epsilon \sum_{i=1}^m w_i \nabla L(\theta)^\top \delta_i + \frac{\epsilon^2}{2}\sum_{i=1}^m w_i \delta_i^\top A(\theta)\delta_i + \frac{\epsilon^2}{2}\sum_{ij=1}^m w_i w_j \delta_i^\top B(\theta)\delta_j + \mathcal{O}(\epsilon^3)$$

$$= \epsilon \nabla L(\theta)^\top \bar{\delta} + \frac{\epsilon^2}{2}\sum_{i=1}^m w_i \delta_i^\top A(\theta)\delta_i + \frac{\epsilon^2}{2}\bar{\delta}^\top B(\theta)\bar{\delta} + \mathcal{O}(\epsilon^3)$$

$$= \epsilon \nabla L(\theta)^\top \bar{\delta} + \frac{\epsilon^2}{2}\bar{\delta}^\top (A(\theta) + B(\theta))\bar{\delta} + \frac{\epsilon^2}{2}\sum_{i=1}^m (\delta_i^\top A \delta_i - \bar{\delta}^\top A \bar{\delta}) + \mathcal{O}(\epsilon^3)$$

$$= \epsilon \nabla L(\theta)^\top \bar{\delta} + \frac{\epsilon^2}{2}\bar{\delta}^\top \nabla^2 L(\theta)\bar{\delta} + \frac{\epsilon^2}{2}\sum_{i=1}^m (\delta_i - \bar{\delta})^\top A(\theta)(\delta_i - \bar{\delta}) + \mathcal{O}(\epsilon^3).$$

This completes the proof. $\qquad\square$

***Proof of Theorem 2.3.***  Recall that

$$\Pi(\boldsymbol{\delta}, \boldsymbol{w}; \theta) = \frac{\epsilon^2}{2}\sum_{i=1}^m w_i \delta_i^\top S(\theta)\delta_i,$$

with $\sum_i w_i = 1$, $w_i \geq 0$ and $\|\delta_i\| = 1$. Since $\delta_i^\top S(\theta)\delta_i \geq \lambda_{min}(S(\theta)) \|\delta_i\|^2 = \lambda_{min}(S(\theta))$, it is obvious that

$$II(\boldsymbol{\delta}, \boldsymbol{w}; \theta) = \frac{\epsilon^2}{2} \sum_{i=1}^m w_i \delta_i^\top S(\theta)\delta_i \geq \frac{\epsilon^2}{2} \sum_{i=1}^m w_i \lambda_{min}(S(\theta)) = \frac{\epsilon^2}{2} \lambda_{min}(S(\theta)).$$

On the other hand, this lower bound is achieved by setting $m = 2$, $w_1 = w_2 = 1/2$ and $\delta_1 = -\delta_2 = v_{min}(S(\theta))$. This completes the proof. $\qquad\square$

***Proof of Theorem 2.4.*** Step 1: We first consider the case with no average displacement, that is, $\mu^{[\ell]} = 0$. In this case, Lemma A.1 below gives

$$\mathcal{L}(\boldsymbol{\theta}^{[1:n]}, \boldsymbol{w}^{[1:n]}) = L(\theta^{[1:n]}) + \sum_{\ell=1}^n \left( \mathcal{L}(\boldsymbol{\theta}_\ell^{[1:n]}, \boldsymbol{w}^{[1:n]}) - L(\theta^{[1:n]}) \right) + \mathcal{O}(\epsilon^3), \qquad (10)$$

where $\boldsymbol{\theta}_\ell^{[1:n]}$ denotes the augmented parameters obtained when we only split the $\ell$-th neuron, while keeping all the neurons unchanged. Applying Theorem 2.2, we have for each $\ell$,

$$\mathcal{L}(\boldsymbol{\theta}_\ell^{[1:n]}, \boldsymbol{w}^{[1:n]}) - L(\theta^{[1:n]}) = \frac{\epsilon^2}{2} II_\ell(\boldsymbol{\delta}^{[\ell]}, \boldsymbol{w}^{[\ell]}; \theta^{[1:n]}) + \mathcal{O}(\epsilon^3).$$

Combining this with (10) yields the result.

Step 2: We now consider the more general case when $\mu^{[1:n]} \neq 0$. Let $\tilde{\theta}^{[1:n]} = \theta^{[1:n]} + \epsilon\mu^{[1:n]}$. Applying the result above on $\tilde{\theta}^{[1:n]}$, we have

$$\mathcal{L}\left(\boldsymbol{\theta}^{[1:n]}, \boldsymbol{w}^{[1:n]}\right) = L\left(\tilde{\theta}^{[1:n]}\right) + \frac{\epsilon^2}{2} D\left(\tilde{\theta}^{[1:n]}\right) + \mathcal{O}(\epsilon^3)$$

where $D\left(\tilde{\theta}^{[1:n]}\right) := \sum_{\ell=1}^n II_\ell(\boldsymbol{\delta}^{[\ell]}, \boldsymbol{w}^{[\ell]}; \tilde{\theta}^{[1:n]})$. Therefore,

$$\begin{aligned}
\mathcal{L}\left(\boldsymbol{\theta}^{[1:n]}, \boldsymbol{w}^{[1:n]}\right) &= L\left(\tilde{\theta}^{[1:n]}\right) + \frac{\epsilon^2}{2} D(\tilde{\theta}^{[1:n]}) + \mathcal{O}(\epsilon^3) \\
&= L\left(\tilde{\theta}^{[1:n]}\right) + \frac{\epsilon^2}{2} D(\theta^{[1:n]}) + \frac{\epsilon^2}{2}(D(\tilde{\theta}^{[1:n]}) - D(\theta^{[1:n]})) + \mathcal{O}(\epsilon^3) \\
&= L\left(\tilde{\theta}^{[1:n]}\right) + \frac{\epsilon^2}{2} D(\theta^{[1:n]}) + \mathcal{O}(\epsilon^3) \qquad \textcolor{magenta}{\text{//because } \theta^{[1:n]} - \tilde{\theta}^{[1:n]} = \mathcal{O}(\epsilon)} \\
&= L\left(\theta^{[1:n]} + \epsilon\mu^{[1:n]}\right) + \frac{\epsilon^2}{2} D(\theta^{[1:n]}) + \mathcal{O}(\epsilon^3),
\end{aligned}$$

where $D\left(\theta^{[1:n]}\right) := \sum_{\ell=1}^n II_\ell(\boldsymbol{\delta}^{[\ell]}, \boldsymbol{w}^{[\ell]}; \theta^{[1:n]})$. This completes the proof. $\qquad\square$

**Lemma A.1.** *Let $\theta^{[1:n]}$ be the parameters of $n$ neurons. Recall that we assume $\theta^{[\ell]}$ is split into $m_\ell$ off-springs with parameters $\boldsymbol{\theta}^{[\ell]} = \{\theta_i^{[\ell]}\}_{i=1}^{m_\ell}$ and weights $\boldsymbol{w}^{[\ell]} = \{w_i^{[\ell]}\}_{i=1}^{m_\ell}$, which satisfies $\sum_{i=1}^{m_\ell} w_i^{[\ell]} = 1$. Let $\theta_i^{[\ell]} = \theta^{[\ell]} + \epsilon\delta_i^{[\ell]}$, where $\delta_i^{[\ell]}$ is the perturbation on the $i$-th off-spring of the $\ell$-th neuron. Assume $\bar{\delta}^{[\ell]} := \sum_{i=1}^{m_\ell} w_i^{[\ell]} \delta_i^{[\ell]} = 0$, that is, the average displacement of all the neurons is zero.*

*Denote by $\boldsymbol{\theta}_\ell^{[1:n]}$ the augmented parameters we obtained by only splitting the $\ell$-th neuron while keeping all the other neurons unchanged, that is, we have $\theta_{\ell,i}^{[\ell]} = \theta^{[\ell]} + \epsilon\delta_i^{[\ell]}$ for $i = 1, \ldots, m_\ell$, and $\theta_{\ell,i}^{[\ell']} = \theta^{[\ell']}$ for all $\ell' \neq \ell$ and $i = 1, \ldots, m_{\ell'}$. Assume the third order derivatives of $\mathcal{L}(\boldsymbol{\theta}^{[1:n]}, \boldsymbol{w}^{[1:n]})$ are bounded. We have*

$$\mathcal{L}(\boldsymbol{\theta}^{[1:n]}, \boldsymbol{w}^{[1:n]}) = L(\theta^{[1:n]}) + \sum_{\ell=1}^n \left( \mathcal{L}(\boldsymbol{\theta}_\ell^{[1:n]}, \boldsymbol{w}^{[1:n]}) - L(\theta^{[1:n]}) \right) + \mathcal{O}(\epsilon^3).$$

*Proof.* Define

$$F := \left( \mathcal{L}(\boldsymbol{\theta}^{[1:n]}, \boldsymbol{w}^{[1:n]}) - L(\theta^{[1:n]}) \right) - \sum_{\ell=1}^{n} \left( \mathcal{L}(\boldsymbol{\theta}_\ell^{[1:n]}, \boldsymbol{w}^{[1:n]}) - L(\theta^{[1:n]}) \right).$$

By Taylor expansion,

$$F = \epsilon \nabla_\epsilon F\big|_{\epsilon=0} + \frac{\epsilon^2}{2} \nabla_{\epsilon\epsilon} F\big|_{\epsilon=0} + \mathcal{O}(\epsilon^3).$$

It is obvious to see that the first order derivation $\nabla_\epsilon F\big|_{\epsilon=0}$ equals zero because of the correction terms. Specifically,

$$\nabla_\epsilon F\big|_{\epsilon=0} = \sum_{\ell=1}^{n} \sum_{i=1}^{m_\ell} \nabla_{\theta_i^{[\ell]}} \mathcal{L}(\boldsymbol{\theta}^{[1:n]}, \boldsymbol{w}^{[1:n]})^\top \delta_i^{[\ell]}\bigg|_{\epsilon=0} - \sum_{\ell=1}^{n} \sum_{i=1}^{m_\ell} \nabla_{\theta_i^{[\ell]}} \mathcal{L}(\boldsymbol{\theta}^{[1:n]}, \boldsymbol{w}^{[1:n]})^\top \delta_i^{[\ell]}\bigg|_{\epsilon=0} = 0.$$

For the second order derivation, define

$$A_{\ell,\ell'} = \nabla_{\theta^{[\ell]}\theta^{[\ell']}} L(\theta^{[1:n]}).$$

For any $\ell \neq \ell'$, we have from (5) and (6) that

$$\nabla_{\theta_i^{[\ell]}\theta_{i'}^{[\ell']}} \mathcal{L}(\boldsymbol{\theta}^{[1:n]}, \boldsymbol{w}^{[1:n]})\bigg|_{\epsilon=0} = w_i^{[\ell]} w_{i'}^{[\ell']} \nabla_{\theta^{[\ell]}\theta^{[\ell']}} L(\theta^{[1:n]}) = w_i^{[\ell]} w_{i'}^{[\ell']} A_{\ell,\ell'}.$$

Therefore, we have

$$\begin{aligned}
\nabla_{\epsilon\epsilon} F\big|_{\epsilon=0} &= \sum_{\ell \neq \ell'} \sum_{i=1}^{m_\ell} \sum_{i'=1}^{m_{\ell'}} (\delta_i^{[\ell]})^\top \nabla_{\theta_i^{[\ell]}\theta_{i'}^{[\ell']}} \mathcal{L}(\boldsymbol{\theta}^{[1:n]}, \boldsymbol{w}^{[1:n]})\bigg|_{\epsilon=0} \delta_{i'}^{[\ell']} \\
&= \sum_{\ell \neq \ell'} \sum_{i=1}^{m_\ell} \sum_{i'=1}^{m_{\ell'}} w_i^{[\ell]} w_{i'}^{[\ell']} (\delta_i^{[\ell]})^\top A_{\ell,\ell'} \delta_{i'}^{[\ell']} \\
&= \sum_{\ell \neq \ell'} (\bar{\delta}^{[\ell]})^\top A_{\ell,\ell'} \bar{\delta}^{[\ell']} \\
&= 0 \qquad \text{//because } \bar{\delta}^{[\ell]} = 0,
\end{aligned}$$

where $\nabla_{\epsilon\epsilon} F\big|_{\epsilon=0}$ only involves cross derivatives $\nabla_{\theta_i^{[\ell]}\theta_{i'}^{[\ell']}} \mathcal{L}(\boldsymbol{\theta}^{[1:n]}, \boldsymbol{w}^{[1:n]})$ with $\ell \neq \ell'$, because all the terms with $\ell = \ell'$ are cancelled due to the correction terms. $\qquad\square$

## A.2  Proofs of $\infty$-Wasserstein Steepest Descent

Recall that $p$-Wasserstein distance is

$$W_p(\rho, \rho') = \inf_{\gamma \in \Pi(\rho,\rho')} \mathbb{E}_{(\theta,\theta')\sim\gamma}[\|\theta - \theta'\|^p]^{1/p}.$$

When $p \to +\infty$, we obtain $\infty$-Wasserstein distance,

$$W_\infty(\rho, \rho') = \inf_{\gamma \in \Pi(\rho,\rho')} \operatorname*{esssup}_{(\theta,\theta')\sim\gamma} \|\theta - \theta'\|, \tag{11}$$

where esssup denotes essential supremum; it is the minimum value $c$ with $\gamma(\|\theta - \theta'\| \geq c) = 0$.

In the proof, we denote by $\gamma_{\rho,\rho'}$ an optimal solution of $\gamma$ in (11), that is,

$$\gamma_{\rho,\rho'} \in \arg \inf_{\gamma \in \Pi(\rho,\rho')} \operatorname*{esssup}_{(\theta,\theta')\sim\gamma} \|\theta - \theta'\|.$$

$\gamma_{\rho,\rho'}$ is called an $\infty$-Wasserstein optimal coupling of $\rho$ and $\rho'$. Denote by $\mu_{\rho,\rho'}(\theta)$ and $\Sigma_{\rho,\rho'}(\theta)$ the mean and covariance matrix of $(\theta' - \theta)$ under $\gamma_{\rho,\rho'}$, conditional on $\theta$, that is,

$$\mu_{\rho,\rho'}(\theta) = \mathbb{E}_{\gamma_{\rho,\rho'}}[(\theta' - \theta) \mid \theta] \qquad\qquad \Sigma_{\rho,\rho'}(\theta) = \operatorname{cov}_{\gamma_{\rho,\rho'}}[(\theta' - \theta) \mid \theta].$$

It is natural to expect that we can upper bound the magnitude of both $\mu_{\rho,\rho'}(\theta)$ and $\Sigma_{\rho,\rho'}(\theta)$ by the $\infty$-Wasserstein distance.

**Lemma A.2.** *Following the definition above, we have*

$$\|\mu_{\rho,\rho'}(\theta)\| \leq W_\infty(\rho, \rho'), \qquad\qquad \lambda_{max}(\Sigma_{\rho,\rho'}(\theta)) \leq W_\infty(\rho, \rho')^2,$$

*almost surely for $\theta \sim \rho$.*

*Proof.* We have

$$\|\mu_{\rho,\rho'}(\theta)\| \leq \operatorname*{esssup}_{\gamma_{\rho,\rho'}} \|\theta - \theta'\| = W_\infty(\rho, \rho'),$$

almost surely for $\theta \sim \rho$. And

$$\begin{aligned}
\lambda_{max}(\Sigma_{\rho,\rho'}(\theta)) &= \max_{\|v\|=1} \operatorname{var}_{\gamma_{\rho,\rho'}}\left[ v^\top (\theta' - \theta) \mid \theta \right] \\
&\leq \max_{\|v\|=1} \mathbb{E}_{\gamma_{\rho,\rho'}}\left[ \left( v^\top (\theta' - \theta) \right)^2 \mid \theta \right] \\
&\leq \operatorname*{esssup}_{\gamma_{\rho,\rho'}} \|\theta - \theta'\|^2 \\
&= W_\infty(\rho, \rho')^2.
\end{aligned}$$

$\square$

**Theorem A.3.** *Define $G_\rho(\theta) = \mathbb{E}_{x\sim\mathcal{D}}\left[\nabla\Phi\left(\mathbb{E}_\rho[\sigma(\theta, x)]\right)\nabla\sigma(\theta, x)\right]$. For two distributions $\rho$ and $\rho'$ and their $\infty$-Wasserstein optimal coupling $\gamma_{\rho,\rho'}$. We have*

$$\mathcal{L}[\rho'] = \mathcal{L}[\rho] + \mathbb{E}_{\theta\sim\rho}\left[G_\rho(\theta)^\top \mu_{\rho,\rho'}(\theta)\right] + \mathcal{O}((\mathbb{D}_\infty(\rho, \rho')^2)). \tag{12}$$

*Proof.* We write $\gamma = \gamma_{\rho,\rho'}$ for convenience. Denote by $\nabla\sigma(\theta, x) = \nabla_\theta\sigma(\theta, x)$ and $\nabla^2\sigma(\theta, x) = \nabla^2_{\theta\theta}\sigma(\theta, x)$ the first and second order derivatives of $\sigma$ in terms of its first variable.

For $(\theta, \theta') \sim \gamma$, introduce $\theta_\eta = \eta\theta' + (1-\eta)\theta$, whose distribution is denoted by $\rho_\eta$. We have $\rho_0 = \rho$ and $\rho_1 = \rho'$. Taking Taylor expansion of $\mathcal{L}[\rho_\eta]$ w.r.t. $\eta$, we have

$$\mathcal{L}[\rho'] = \mathcal{L}[\rho] + \nabla_\eta\mathcal{L}[\rho_\eta]\Big|_{\eta=0} + \frac{1}{2}\nabla^2_{\eta\eta}\mathcal{L}[\rho_\eta]\Big|_{\eta=\xi},$$

where $\xi$ is a number between 0 and 1. We just need to calculate the derivatives. For the first order derivative, we have

$$\begin{aligned}
\nabla_\eta\mathcal{L}[\rho_\eta]\Big|_{\eta=0} &= \nabla_\eta\mathbb{E}_{x\sim\mathcal{D}}\left[\Phi\left(\mathbb{E}_\gamma[\sigma(\eta\theta' + (1-\eta)\theta, x)]\right)\right]\Big|_{\eta=0} \\
&= \mathbb{E}_{x\sim\mathcal{D}}\left[\Phi'\left(\mathbb{E}_\gamma[\sigma(\theta_\eta, x)]\right)\mathbb{E}_\gamma[\nabla\sigma(\theta_\eta, x)^\top(\theta' - \theta)]\right]\Big|_{\eta=0} \\
&= \mathbb{E}_{x\sim\mathcal{D}}\left[\Phi'\left(\mathbb{E}_\gamma[\sigma(\theta, x)]\right)\mathbb{E}_\gamma[\nabla\sigma(\theta, x)^\top(\theta' - \theta)]\right] \\
&= \mathbb{E}_\gamma[G_\rho(\theta)^\top(\theta' - \theta)] \\
&= \mathbb{E}_\rho\left[G_\rho(\theta)^\top\mu_{\rho,\rho'}(\theta)\right],
\end{aligned}$$

where we used the derivation of $G_\rho(\theta)$.

For the second order derivative, we have

$$\begin{aligned}
\nabla^2_{\eta\eta}\mathcal{L}[\rho_\eta]\Big|_{\eta=\xi} &= \nabla_\eta(\nabla_\eta\mathcal{L}[\rho_\eta])\Big|_{\eta=\xi} \\
&= \nabla_\eta\mathbb{E}_{x\sim\mathcal{D}}\left[\Phi'\left(\mathbb{E}_\gamma[\sigma(\theta_\eta, x)]\right)\mathbb{E}_\gamma[\nabla\sigma(\theta_\eta, x)^\top(\theta' - \theta)]\right]\Big|_{\eta=\xi} \\
&= \mathbb{E}_{x\sim\mathcal{D}}\left[\Phi''\left(\mathbb{E}_\gamma[\sigma(\theta_\eta, x)]\right)\left(\mathbb{E}_\gamma[\nabla\sigma(\theta_\eta, x)(\theta' - \theta)]\right)^2\right] \\
&\quad + \mathbb{E}_{x\sim\mathcal{D}}\left[\Phi'\left(\mathbb{E}_\gamma[\sigma(\theta_\eta, x)]\right)\mathbb{E}_\gamma[(\theta' - \theta)^\top\nabla^2\sigma(\theta_\eta, x)(\theta' - \theta)]\right]\Big|_{\eta=\xi} \\
&= \mathbb{E}_\gamma[(\theta' - \theta)^\top T_\rho(\theta_\xi)(\theta' - \theta)] + \mathbb{E}_\gamma[(\theta' - \theta)^\top S_\rho(\theta_\xi)(\theta' - \theta)] \\
&= \mathbb{E}_\gamma[(\theta' - \theta)^\top(T_\rho(\theta_\xi) + S_\rho(\theta_\xi))(\theta' - \theta)]
\end{aligned}$$

where we define $T_\rho(\theta_\xi) := \mathbb{E}_{x\sim\mathcal{D}}\left[\Phi''\left(\mathbb{E}_\gamma[\sigma(\theta_\xi, x)]\right)\nabla\sigma(\theta_\xi, x)^{\otimes 2}\right]$. Denote by $\lambda_* :=$ $\sup_{\xi\in[0,1]}\lambda_{max}(T_\rho(\theta_\xi) + S_\rho(\theta_\xi))$. We have

$$
\begin{aligned}
\nabla^2_{\eta\eta}\mathcal{L}[\rho_\eta]\bigg|_{\eta=\xi} &\le \lambda_*\mathbb{E}_\gamma\left[\|\theta' - \theta\|^2\right] \\
&= \mathcal{O}\left(\mathbb{E}_\gamma\left[\|\theta' - \theta\|^2\right]\right) \\
&= \mathcal{O}\left(\left[\operatorname*{esssup}_\gamma\|\theta' - \theta\|\right]^2\right) \\
&= \mathcal{O}(\mathbb{D}_\infty(\rho, \rho')^2).
\end{aligned}
$$

This completes the proof. $\qquad\square$

**Theorem A.4.** *For two distributions $\rho$ and $\rho'$, denote by $\gamma_{\rho,\rho'}$ their $\infty$-Wasserstein optimal coupling, and $\mu_{\rho,\rho'}(\theta)$ and $\Sigma_{\rho,\rho'}(\theta)$ the mean and covariance matrix of $(\theta' - \theta)$ under $\gamma_{\rho,\rho'}$, conditional on $\theta$, respectively. Denote by $(I + \mu_{\rho,\rho'})\sharp\rho$ the distribution of $\theta + \mu_{\rho,\rho'}(\theta)$ when $\theta \sim \rho$. We have*

$$
\mathcal{L}[\rho'] = \mathcal{L}\left[(I + \mu_{\rho,\rho'})\sharp\rho\right] + \mathbb{E}_{\theta\sim\rho}\left[\frac{1}{2}\operatorname{tr}\left(S_\rho(\theta)^\top\Sigma_{\rho,\rho'}(\theta)\right)\right] + \mathcal{O}((\mathbb{D}_\infty(\rho, \rho'))^3) \tag{13}
$$

*where $S_\rho(\theta) = \mathbb{E}_{x\sim\mathcal{D}}\left[\Phi'(f_\rho(x))\nabla^2_{\theta\theta}\sigma(\theta, x)\right]$. The first and second terms capture the effect of displacement and splitting, respectively.*

***Proof of Theorem A.4.*** We use $\gamma := \gamma_{\rho,\rho'}$ for notation convenience. Denote by $\tilde\theta = \theta + \mu_{\rho,\rho'}(\theta)$ and $\tilde\rho = (I + \mu_{\rho,\rho'})\sharp\rho$ the distribution of $\tilde\theta$ when $\theta \sim \rho$. Recall that for $(\theta, \theta') \sim \gamma$, we have

$$
\mathbb{E}_\gamma\left[\theta' - \tilde\theta \,\middle|\, \theta\right] = \mathbb{E}_\gamma[\theta' - \theta - \mu_{\rho,\rho'}(\theta) \mid \theta] = 0, \tag{14}
$$

$$
\Sigma_{\rho,\rho'}(\theta) = \mathbb{E}_\gamma\left[(\theta' - \tilde\theta)(\theta' - \tilde\theta)^\top \,\middle|\, \theta\right]. \tag{15}
$$

Introduce $\theta_\eta = \eta\theta' + (1 - \eta)\tilde\theta$. Denote by $\rho_\eta$ the distribution of $\theta_\eta$. This gives $\rho' = \rho_1$ and $\tilde\rho = \rho_0$. We have

$$
\mathcal{L}[\rho'] = \mathcal{L}[\tilde\rho] + \nabla_\eta\mathcal{L}[\rho_\eta]\bigg|_{\eta=0} + \frac{1}{2}\nabla^2_{\eta\eta}\mathcal{L}[\rho_\eta]\bigg|_{\eta=0} + \frac{1}{6}\nabla^3_{\eta\eta\eta}\mathcal{L}[\rho_\eta]\bigg|_{\eta=\xi},
$$

where $\xi$ is a number between 0 and 1. We just need to evaluate these derivatives. For the first order derivative, we have

$$
\begin{aligned}
\nabla_\eta\mathcal{L}[\rho_\eta]\bigg|_{\eta=0} &= \nabla_\eta\mathbb{E}_{x\sim\mathcal{D}}\left[\Phi\left(\mathbb{E}_\gamma[\sigma(\eta\theta' + (1 - \eta)\tilde\theta, x)]\right)\right]\bigg|_{\eta=0} \\
&= \mathbb{E}_{x\sim\mathcal{D}}\left[\Phi'\left(\mathbb{E}_\gamma[\sigma(\theta_\eta, x)]\right)\mathbb{E}_\gamma[\nabla\sigma(\theta_\eta, x)^\top(\theta' - \tilde\theta)]\right]\bigg|_{\eta=0} \\
&= \mathbb{E}_{x\sim\mathcal{D}}\left[\Phi'\left(\mathbb{E}_\gamma[\sigma(\tilde\theta, x)]\right)\mathbb{E}_\gamma[\nabla\sigma(\tilde\theta, x)^\top(\theta' - \tilde\theta)]\right] \\
&= 0,
\end{aligned}
$$

where the last step uses (14). Here $\nabla\sigma$ denote the derivatives w.r.t. its first variables.

For the second order derivative, we have

$$
\begin{aligned}
\nabla^2_{\eta\eta}\mathcal{L}[\rho_\eta]\Big|_{\eta=0} &= \nabla_\eta(\nabla_\eta\mathcal{L}[\rho_\eta])\Big|_{\eta=0} \\
&= \nabla_\eta\mathbb{E}_{x\sim\mathcal{D}}\left[\Phi'\left(\mathbb{E}_\gamma[\sigma(\theta_\eta,x)]\right)\mathbb{E}_\gamma[\nabla\sigma(\theta_\eta,x)^\top(\theta'-\tilde{\theta})]\right]\Big|_{\eta=0} \\
&= \mathbb{E}_{x\sim\mathcal{D}}\left[\Phi''\left(\mathbb{E}_\gamma[\sigma(\theta_\eta,x)]\right)(\mathbb{E}_\gamma[\nabla\sigma(\theta_\eta,x)(\theta'-\tilde{\theta})])^2\right] \\
&\quad + \mathbb{E}_{x\sim\mathcal{D}}\left[\Phi'\left(\mathbb{E}_\gamma[\sigma(\theta_\eta,x)]\right)\mathbb{E}_\gamma[(\theta'-\tilde{\theta})^\top\nabla^2\sigma(\theta_\eta,x)(\theta'-\tilde{\theta})]\right]\Big|_{\eta=0} \quad\quad (16) \\
&= 0 + \mathbb{E}_\gamma[(\theta'-\tilde{\theta})^\top S_\rho(\theta)(\theta'-\tilde{\theta})] \\
&= \mathbb{E}_{\theta\sim\rho}[\text{tr}(S_\rho(\theta)\Sigma_{\rho,\rho'}(\theta))].
\end{aligned}
$$

Further, we can show that $\nabla^3_{\eta\eta\eta}\mathcal{L}[\rho_\eta]\Big|_{\eta=\xi} = \mathcal{O}(\mathbb{D}_\infty(\rho,\rho')^3)$, since when taking the third gradient, all the terms of the derivative are bounded by $\|\theta-\theta'\|^3$. Specifically, taking the derivative of the form of $\nabla^2_{\eta\eta}\mathcal{L}[\rho_\eta]$ in (16) gives

$$
\begin{aligned}
&\nabla^3_{\eta\eta\eta}\mathcal{L}[\rho_\eta]\Big|_{\eta=\xi} \\
&= \mathbb{E}_{x\sim\mathcal{D}}\left[\Phi'''\left(\mathbb{E}_\gamma[\sigma(\theta_\xi,x)]\right)(\mathbb{E}_\gamma[\nabla\sigma(\theta_\xi,x)(\theta'-\tilde{\theta})])^3\right] \\
&\quad + 3\mathbb{E}_{x\sim\mathcal{D}}\left[\Phi''\left(\mathbb{E}_\gamma[\sigma(\theta_\xi,x)]\right)\mathbb{E}_\gamma[\nabla\sigma(\theta_\xi,x)(\theta'-\tilde{\theta})]\mathbb{E}_\gamma[(\theta'-\tilde{\theta})^\top\nabla^2\sigma(\theta_\xi,x)(\theta'-\tilde{\theta})]\right] \\
&\quad + \mathbb{E}_{x\sim\mathcal{D}}\left[\Phi'\left(\mathbb{E}_\gamma[\sigma(\theta_\xi,x)]\right)\mathbb{E}_\gamma[\langle\nabla^3\sigma(\theta_\xi,x),\ (\theta'-\tilde{\theta})^{\otimes3}\rangle]\right] \\
&= \mathcal{O}\left(\underset{(\theta,\theta')\sim\rho}{\text{esssup}}\left\|\theta'-\tilde{\theta}\right\|^3\right) \\
&= \mathcal{O}(\mathbb{D}_\infty(\rho,\rho')^3).
\end{aligned}
$$

Here we use the notation $\langle A,\ v^{\otimes3}\rangle = \sum_{ijk=1}^d A_{ijk}v_iv_jv_k$. This completes the proof. $\square$

***Proof of Theorem 2.5.*** Following Theorem A.3, we have

$$
\Delta^*(\rho,\epsilon) = \min_{\rho'}\left\{\mathbb{E}_{\theta\sim\rho}\left[G_\rho(\theta)^\top\mu_{\rho,\rho'}(\theta)\right]:\quad \mathbb{D}_\infty(\rho,\rho')\le\epsilon\right\} + \mathcal{O}(\epsilon^2).
$$

For $\mathbb{D}_\infty(\rho,\rho')\le\epsilon$, we must have $\|\mu_{\rho,\rho'}\|\le\epsilon$, and hence $\mathbb{E}_{(\theta,\theta')\sim\gamma_{\rho,\rho'}}\left[G_\rho(\theta)^\top\mu_{\rho,\rho'}(\theta)\right]\ge -\epsilon\mathbb{E}_\rho[\|G_\rho(\theta)\|]$ by Cauchy–Schwarz inequality. On the other hand, this minimum is achieved when $\mu_{\rho,\rho'} = -\epsilon G_\rho(\theta)/\|G_\rho(\theta)\|$. The only distribution $\rho'$ that satisfies this condition is $\rho' = (I - \epsilon G_\rho(\theta)/\|G_\rho(\theta)\|)\sharp\rho$. This proves Theorem 2.5a.

For Theorem 2.5b, we need to use the result in Theorem A.4, which yields, in the case of stable local optima, that

$$
\Delta^*(\rho,\epsilon) = \min_{\rho'}\left\{\mathbb{E}_{\theta\sim\rho}\left[\frac{1}{2}\text{tr}\big(S_\rho(\theta)^\top\Sigma_{\rho,\rho'}(\theta)\big)\right]:\quad \mathbb{D}_\infty(\rho,\rho')\le\epsilon\right\} + \mathcal{O}(\epsilon^3).
$$

Similar to the argument above, the minima should satisfy $\Sigma_{\rho,\rho'}(\theta)\propto v_{min}v_{min}^\top$, where $v_{min}$ is the eigenvector of $S_\rho(\theta)$ associated with its minimum eigenvalue. This corresponds to splitting $\theta$ into two copies with each weights with parameter $\theta\pm\epsilon v_{min}$ when $\lambda_{min}<0$, or keep $\rho$ unchanged when $\lambda_{min}>0$. $\square$

# B Experimental Settings and Additional Results

## B.1 Two-Layer RBF Neural network

We consider fitting a simple radial basis function (RBF) neural network of form

$$f(x) = \sum_{i=1}^{m} \sigma(\theta_i, x), \qquad \sigma(\theta, x) := \theta_{i,3} \times \exp\left(-\frac{1}{2}(\theta_{i,1}x + \theta_{i,2})^2\right),$$

where $x \in \mathbb{R}$ and $\theta_i = [\theta_{i,1}, \theta_{i,2}, \theta_{i,3}]^\top \in \mathbb{R}^3$. For the ground truth, we set $m = 15$ and sample the true values of parameters $\{\theta_i\}$ from $\mathcal{N}(0, 3)$, yielding the light blue curves shown in Figure 5. We generate a training data set $\mathcal{D} := \{x_i, y_i\}_{i=1}^{1000}$ by drawing $x_i$ from $\mathrm{Uniform}[-5, 5]$ and set $y_i = f_*(x_i)$ without noise, where $f_*$ denotes the true network we sampled. The network is trained by minimizing the mean square loss:

$$\min_f \mathbb{E}_{x \sim \mathcal{D}} \left[ (f_*(x) - f(x))^2 \right].$$

Mapping to (2), we have $\Phi(f) = (f_* - f)^2$. We learn the function using our splitting method and other progressive training baselines, all starting from $m = 1$ neuron. We add one additional neuron in each splitting/growing phase for all the methods. The parametric descent phase is performed using typical stochastic gradient descent until convergence. We stop the splitting process at $m = 8$ for all the methods. Figure 5 shows curves learned by different methods with $m = 3$ and $m = 8$ neurons, respectively. Our method yields better approximation.

(a) $m = 3$        (b) $m = 8$

Figure 5: Results on the toy RBF neural network.

## B.2   Learning Interpretable Neural Network

We provide more details on learning the interpretable neural network.

**Setting**   We adopt the interpretable neural architecture proposed in Li et al. (2018) as our testbed. Unlike standard black-box neural networks, this architecture contains a special prototype layer in the classifier, which includes a set of prototype neurons that are enforced to encode to realistic images for promoting interpretability. In this model, each input image $x$ is first mapped to a lower-dimensional representation based on its distance $\|\theta - e(x)\|$ with a set of prototype vectors, where $\theta \in \mathbb{R}^{40}$ represents a prototype vector and $e(x)$ is an encoder function. The prototype vectors are enforced to be interpretable in that they can be decoded to some realistic images; this is achieved in Li et al. (2018) by introducing a regularization term that minimizes the minimum square distance between the prototypes and the training data, that is, $\min_i \|\theta - e(x_i)\|$, where $\{x_i\}$ denotes the training dataset.

We apply our method to split the prototype neurons, by treating $\sigma(\theta, x) := \|\theta - e(x)\|$ as the activation function. We use the MNIST dataset in our experiment. We visualize the prototype neurons we learned using the images that they encode, by feeding the prototype vectors $\theta$ into a decoder function jointly trained with the network. We use the same encoder and decoder architectures, as suggested in Li et al. (2018) and refer the reader to Li et al. (2018) for more implementation details. To better understand the splitting dynamics, we start with a small network with just one prototype neuron and gradually add more prototypes via splitting.

We compare our method with two baseline methods, `New Initialization` and `Random Split`, that also progressively grow the prototype layers starting from one prototype neuron. In `New Initialization`, we simply add one new prototype neuron with random initialization at each iteration. In `Random Split`, we randomly pick a prototype neuron to split and split it following its splitting gradient given by our splitting matrix. Figure 6 visualizes the full splitting/growing process of our method and the two baselines. We can see that our splitting method successfully identifies the most ambiguous (and least interpretable) prototype neurons to split at each iteration, and achieves the best final results.

Optimal Split (Ours)

Random Split

New Initialization

Figure 6: Visualizing the growing process of the prototype neurons given by our splitting method and the two baselines.

## B.3 Lightweight Neural Architectures for Image Classification

We describe details of our experiments on learning lightweight deep networks for image classification.

**Dataset and Backbone Networks**   We use the CIFAR-10 benchmark dataset. We adopt a standard data argumentation scheme (mirroring and shifting) that is widely used for this dataset (Liu et al., 2019b, 2017). The input images are normalized using channel means and standard derivations. We use two popular deep neural architectures as our testbed, MobileNet (Howard et al., 2017) and VGG19 (Simonyan & Zisserman, 2015).

**Training Settings**   We treat the filters as the neurons to split for convolutional neural networks. For example, consider a convolutional layer with $n_{out} \times n_{in} \times k \times k$ parameters, where $n_{out}$ denotes the number of output channels and $n_{in}$ the number of input channels and $k$ the filter size. We treat it as $n_{out}$ neurons, and each neuron has a parameter of size $n_{in} \times k \times k$. To apply our methods, we start with a small variant of the MobileNet and VGG19, and gradually grow the network by splitting the (convolutional) neurons with the most negative splitting indexes following Algorithm 1. For MobileNet, we construct the initial network by keeping the size of the first convolution layer as the same (=32) as the original MobileNet and setting the number of depthwise and pointwise channels to be 16. For VGG19, we set the number of channels of the initial network to be 16 for all layers.

For the parametric descent phase, we use stochastic gradient descent with an initial learning rate 0.1 for 160 epochs. The learning rate is divided by 10 at 50% and 75% of the total number of training epochs. We use a weight decay of $10^{-4}$ and a Nesterove momentum of 0.9 without dampening. The batch size is set to be 64. In each splitting phase, we increase the number of channels by a percentage of 30 using our method.

Note that our splitting matrix (see Eq. 7) involves the second-order derivative of the activation function, which is not well defined for ReLU activation. Therefore, we replace the ReLU activation with Softplus to prevent numerical issues in calculating the splitting matrices. We also apply Softplus in the other experiments that contain ReLU activation function in the network.

**Pruning**   We compare with two model pruning algorithms: the batch-normalization-based pruning (Bn-prune) by Liu et al. (2017) and the L1-based pruning (L1-prune) by Li et al. (2017). Bn-prune imposes L1-sparsity on the channel-wise scaling factors in the batch normalization layers during training, and prunes channels with lower scaling factors afterwards. L1-prune removes the filters with weights of small L1-norm in each layer. For both pruning baselines, we use the implementation provided by Liu et al. (2019b). For Bn-prune, we set the sparsity term to be 0.0001 for all the cases. We initial both pruning methods from a full-size backbone network (MobileNet and VGG19) that we trained starting from scratch. After each pruning phase, the parameters of the pruned network are finetuned starting from the previous values using stochastic gradient descent, following the same setting as that we use in splitting steepest descent.

**Finetuning vs. Retraining**   In both the splitting and pruning methods above, the parameters of the split/pruned networks are successively *finetuned* starting from the previous values. In order to test the performance of the network architectures given by both splitting and pruning methods, we test another setting in which we *retrain* the network parameters after each splitting/pruning step, that is, we discard all the parameters of the network, and retrain the whole network starting from a random initialization, under the network structure obtained from splitting or pruning at each iteration. As shown in Figure 3c-d, the results of retraining is comparable with (or better than) the result of successive finetuning in Figure 3a-b, which is consistent with the findings in Liu et al. (2019b).

## B.4 Resource-Efficient Keyword spotting

We apply our methods on the application of keyword spotting. Keyword spotting systems aim to detect a particular set of keywords from a continuous stream of audio, which is typically deployed on a wide range of edge devices with resource constraints.

**Dataset and Training Settings**   We use the Google speech commands benchmark dataset (Warden, 2018) for comparisons. We are interested in the setting that the model size is limited to less than 500K and adopt the optimized architectures with tight resource constraints provided in Zhang et al.

(2017) as our baselines. For fair comparison, we closely follow the experimental settings described in Zhang et al. (2017). We split the dataset into 80/10/10% for training, validation and test, respectively.

We start with a very narrow network and progressively grow it using splitting steepest descent. We build our initial narrow network based on the DS-CNN architecture proposed in Zhang et al. (2017), by reducing the number of channels in each layer to 16. The backbone DS-CNN model consists of one regular convolution layer and five depthwise and pointwise convolution layers (Howard et al., 2017). We refer the reader to Zhang et al. (2017) for more information. At each splitting stage, we increase the number of channels by a percentage of 30% using the approach described in Algorithm 1. We use the same hyper-parameters for training and evaluation as in Zhang et al. (2017).

## B.5 Splitting Steepest Descent for Minimizing MMD

We consider the problem of data compression. Given a large set of data points $\{\theta_i^*\}_{i=1}^N$, we want to find a smaller set of points $\{\theta_i\}_{i=1}^n$, equipped with a set of importance weights $\{w_i\}_{i=1}^n$, to approximate the larger dataset. This problem can be solved by minimizing maximum mean discrepancy (MMD) (Gretton et al., 2012) using conditional gradient method (a.k.a. Frank-Wolfe), an algorithm known as *herding* (Chen et al., 2010; Bach et al., 2012). In this section, we provide additional results on using splitting steepest descent to minimize MMD by progressively introducing new points via splitting.

Denote by $\rho_* = \sum_{i=1}^N \delta_{\theta_i^*}/N$ the empirical distribution of the original dataset, and $\rho = \sum_{i=1}^n w_i \delta_{\theta_i}$ the (weighted) empirical distribution of the compressed data. Let $k(\theta, \theta')$ be a positive definite kernel, which can be represented using a random feature expansion of form

$$k(\theta, \theta') = \mathbb{E}_{x \sim \pi}[\sigma(\theta, x)\sigma(\theta', x)],$$

where $\sigma(\theta, x)$ is a feature map index by an auxiliary variable $x$, and $\pi$ is a distribution on $x$. The $\sigma(\theta, x)$ can be taken to be the cosine function for commonly used kernels such as RBF kernel; see Rahimi & Recht (2007) for more information on random feature expansion. Then the MMD between $\rho$ and $\rho^*$, with kernel $k(\theta, \theta')$, can be written into

$$\begin{aligned} \text{MMD}(\rho, \rho_*) &= \mathbb{E}_{\rho, \rho_*}[k(\theta, \theta') - 2k(\theta, \theta_*') + k(\theta_*, \theta_*')] \\ &= \mathbb{E}_{x \sim \pi}[(\mathbb{E}_{\theta \sim \rho}[\sigma(\theta, x)] - \mathbb{E}_{\theta_* \sim \rho_*}[\sigma(\theta_*, x)])^2], \end{aligned} \tag{17}$$

where $\theta, \theta'$ are i.i.d. drawn from $\rho$ and $\theta_*, \theta_*'$ are i.i.d. drawn from $\rho_*$. The data compression problem can be viewed as minimizing the MMD:

$$\min_\rho \{\mathcal{L}[\rho] := \text{MMD}(\rho, \rho_*)\}.$$

From (17), this minimization can be viewed as performing least square regression on a one-hidden-layer neural network $f_\rho(x) = \mathbb{E}_{\theta \sim \rho}[\sigma(\theta, x)]$, where each data point $\theta_i$ is viewed as a neuron. Therefore, splitting steepest descent can be applied to minimize the loss function. This allows us to start with a small number of data points (neurons), and gradually increase the number of points by splitting. The splitting matrix of $\mathcal{L}[\rho]$ is

$$\begin{aligned} S_\rho(\theta) &= 2\mathbb{E}_{x \sim \pi}\left[(\mathbb{E}_{\theta' \sim \rho}[\sigma(\theta', x)] - \mathbb{E}_{\theta_* \sim \rho_*}[\sigma(\theta_*, x)])\nabla_{\theta\theta}^2 \sigma(\theta, x)\right] \\ &= 2\mathbb{E}_{\theta' \sim \rho, \theta_* \sim \rho_*}\left[\nabla_{\theta\theta}^2 k(\theta, \theta') - \nabla_{\theta\theta}^2 k(\theta, \theta_*)\right]. \end{aligned}$$

We apply splitting steepest descent (`Optimal Split`) in Algorithm 1 starting from a single point (neuron). We compare our method with `Random Split`, `Gradient Boosting` (a.k.a. Frank-Wolfe or herding), `New Initialization`. In `Random Split`, we randomly pick a point to split, and split it following its splitting gradient direction. In `Gradient Boosting`, a new point is introduced greedily at each iteration by minimizing the MMD loss, with all the previous points fixed. In `New Initialization`, a new random point is introduced and co-optimized together with all the previous points at each iteration.

In our experiment, we construct $\rho_*$ by drawing an i.i.d. sample of size $N = 1000$ from a one-dimensional Gaussian mixture model $0.2\mathcal{N}(-2, 0.5) + 0.3\mathcal{N}(1., 0.5) + 0.5\mathcal{N}(3, 0.5)$ as ground truth. We initialize all the methods from a same point drawn from $\text{Uniform}[-5, -3]$, and add a new point in each splitting/growing phase. The parametric descent phase is performed using the adagrad optimizer with a constant learning rate $0.01$ for all the methods.

Figure 7 plots the training dynamics of all the methods. The size of each dot represents the particle weight. Note that in `Optimal Split` and `Random Split`, each off-spring shares half of the weights of their parent points, but in `New Initialization` and `Gradient Boosting`, all the points evenly divide the weights all the time.

Figure 7: MMD minimization for data compression using different progressive optimization methods.

Figure 8 shows the training iterations vs. the training loss (logarithm of MMD) of our method and the baseline approaches. As we can see from Figure 8, our method yields the lowest training loss in general. The kicks of `New Initialization` and `Gradient Boosting` are resulted from re-weighting all particles after introducing new particles.

Figure 8: Lose curve of different methods for MMD minimization.