[Reviews · NeurIPS 2019]

Reviewer 1



As far as I know, this is the first paper I have met on progressive training of NN with both nice theoretical backup and empirical performance. I did not check all the proofs in the appendix, but they appear reasonable to me. For the experimental part, I think it lacks the training efficiency comparison between splitting GD and existing pruning methods, where in the paper only the accuracy comparison is given.

Reviewer 2



The paper introduces an algorithm for splitting neurons when training neural networks using gradient descent, in order to progressively grow a neural network architecture for a better fit to the data. Some theoretical connections to wasserstein-infinity gradient steps are presented, as well as multiple experiments for learning interpretable or lightweight neural networks. The proposed method is original and interesting, and the experiments suggest that it is promising for various applications. However, the current proposed method seems far from practical in large scale settings, and the presented theoretical results are quite limited: * computationally, the algorithm as presented in Algorithm 1 requires (i) to check if a local optima is reached on the full training loss (and it seems that gradient updates are performed on the full training loss, which is costly), (ii) eigenvector computation on splitting matrices, (iii) dealing with dynamically growing weight matrices. These aspects seem to make the algorithm impractical compared to simply running SGD on an over-parameterized network. * theoretically, the splitting may help escape some local minima by changing the loss landscape, but this only happens when the splitting matrix has negative eigenvalues, which may not necessarily hold in practice. Then, the algorithm may still get stuck (at points that are "splitting stable") and not converge to a smaller loss which may have been reached when training with many more neurons from the beginning (e.g. in the over-parameterized setting of Mei et al. or Chizat and Bach, where it is possible to attain global convergence). Perhaps the authors can give examples where the hessian is positive but S(theta) is negative in order to highlight situations where this approach would be useful. * the links with Wasserstein-infty steepest descent are also unclear: Theorem 2.5(a) suggests that the equivalence is with *normalized* gradient updates (G(theta) / ||G(theta)||), while the algorithm seems to rely on standard gradient updates (corresponding to Wasserstein-2). The claim about the "non-asymptotic" nature of the result is misleading, since it only applies locally on a single update (rather than for global convergence), something which also would hold for the Wasserstein 2 setup even with empirical measures. Minor comments/typos: - L48: inherent -> inherit - L58/59: one would argue that SGD is more of a driving horse than GD for deep learning? - L72: add reference for the O(eps^2) claim? footnote: reference for "happens rarely"? - Algorithm 1, point 1.: more details would be helpful (what updates? how do you check local optimality?) - L179: "often not large in practice": clarify? - L206-208: is this referring to finite p or infinite? what is meant by "no practical difference"? (the number of particles for a good approximation may be very large, thus impractical) ======= after rebuttal ======== I am increasing my score as the proposed method seems more usable in practice than I initially thought, and the theory might be useful for the community. Perhaps the authors could provide more information on the final architectures found by their approach, for instance in the experiments of Figure 3. In particular, I wonder if the final architectures have layers with reasonably similar sizes, in which case the the baseline of gradually increasing the size of all layers equally + retraining from scratch (which doesn't require splitting) might be quite competitive, given the good performance of the 'retrain' strategy.

Reviewer 3



Before beginning, I need to underline that I am not very much familiar with the related work, so my assessment about the novelty of this work is primarily based on the author's own claims in the paper, although I have made some additional reading on the related literature for evaluating this submission. I enjoyed reading the paper. It is technically sound, all claims are supported by both theoretical analysis and empirical experiments. It is clearly written and well organized. Easy to read and understand, even for someone outside of the field. Results of the numerical experiments are very well illustrated. The new approach is expected to have a practical impact, especially for learning lightweight neural architectures. Some minor comments / typos: Figure 1 (a): f(x) is difficult to see even from a colored print-out Line 102: can be derive - can be derived Line 447: Nesterove - Nesterov ========== After author feedback ========== I read the author feedback carefully. During the discussions, I realized that I have misinterpreted some of the reported results as if the proposed approach outperforms the retraining of the final architecture from scratch. As a result, I overrated the practical significance of the proposed approach in my initial comments. To reflect this, I slightly decrease my score. Nevertheless, the proposed approach still has an adequate level of practical interest, and I vote and argue for the acceptance of the manuscript.

[Author Response · NeurIPS 2019]

**[Reviewer #3]** Thank you for your valuable comments and suggestions.

*"Training efficiency comparison"*: 1) Because splitting GD and pruning methods work in a very different fashion, it is
difficult to draw a concrete timing comparison. However, generally speaking, the pruning methods need to start with a
large network whose training cost is high, and is hence very costly when the goal is to learn small-size networks (for
e.g., energy efficiency). In comparison, our splitting strategy only requires to incrementally train small-size networks
and is hence more preferred for learning light-weight neural architectures. Our experiments show that splitting method
can also obtain higher accuracy with the same network size compared with prunning methods; this is in sharp contrast
with the common believe that it is critical to take advantage of the knowledge of the pre-trained large networks.

2) With the gradient descent approximation and auto-differentiation trick shown in Appendix C.3, we can now
implement the eigen-calculation of the splitting matrices very efficiently, with a time complexity comparable to the
regular back-propagation on the same network (see more details in Appendix C.3). We will release our implementation.

3) Because the number of steps of splitting is linear (not exponential) to the size of the final network, our approach is
*much faster* than the existing neural architecture search (NAS) methods that require random or brute force search in an
exponentially large model space. We will add more discussion on the time efficiency in the revision.

**[Reviewer #5]** Thanks for your valuable time and comments. We do respectfully disagree with your rejection based on
concerns on computational cost and convergence, with SGD on over-parametrized networks as the baseline.

*Our work is designed for learning small network structures*: Although SGD on over-parameterized networks have
good empirical and theoretical properties, they yields large and redundant networks that are slow and costly in the
inference phase. The goal of our algorithm is to learn small network architectures that are fast and energy efficient in the
testing phase, for resource constrained settings such as mobile and IoT. Therefore, our method should be compared with
existing methods of the same purpose, including (1) pruning-based methods, which obtain small networks by trimming
or compressing a pre-trained large network, and (2) neural architecture search, which searches for good architectures
starting from scratch by random or black-box optimization (such as evolutionary strategy or policy gradient).

*Our method is fast and practical*: As we elaborate in the response to Reviewer 3, our method has significant advantage
over pruning-based and NAS methods. We believe that our method yields one of the fastest strategy for efficient neural
structure optimization. *We will release our code to demonstrate this after acceptance.* Further comments:

i) We use regular mini-batch gradient descent to train networks with fixed structures. The convergence can be quickly
checked by averaging over a large number of min-batches (and it just takes one epoch of gradient computation even if
we average over the whole dataset). In fact, because the effect of splitting and parametric updates are decoupled as
shown in Eq(4), it is fine to split even when SGD is not strictly converged.

ii) The eigen-computation is fast. First, the cost exact eigen-computation is $O(nd^3)$, where $n$ is the number of neurons
to be split and $d$ the parameters of each neuron. This is not huge for modern machines because $d$ is small compared
with the overall parameter size. For example, a $3 \times 3$ Conv-filter with 64 input channels has $d = 64 \times 3 \times 3 = 576$.
In addition, because we only need to calculate the minimum eigenvalue/eigenvector, we can use the gradient descent
approximation we described in Appendix C.3 to significantly speedup the computation. Our current implementation
allows us to calculate the eigens of all the neurons jointly (without actually expending the matrix $S(\theta)$) with a cost
roughly equivalent to back-propagating on the same network. We will release our implementation.

iii) For dealing with dynamic growing weight matrices, deep learning frameworks such as Pytorch support dynamic
computational graph, which allows us to implement our method easily in practice. See the "rethinking-network-pruning"
repository by github/Eric-mingjie for an example. Once again, we will release our implementation.

*Convergence Guarantees and other*: Our convergence guarantee is better or at least as good as (parametric) SGD
because we can improve the loss of SGD by splitting. Note that all our experiments are examples when parametric
SGD has been stuck at local optimal, but splitting allows us to further decrease the loss (by escaping a "functional"
saddle point). In particular, see the "cliff points" in the Fig 1(d). The normalization in $G(\theta)/||G(\theta)||$ is not substantial
and impacts the step size, which we do not discuss in the work. "Non-asymptotic" refers the point that works like Mei
etal is based on asymptotic of infinite number of neurons, while our method does not. We will use the term "mean field"
to avoid confusion. Further clarification will be provided.

**[Reviewer #6]** Thank you for your valuable comments and suggestions. Our work provides a new framework of neural
structure optimization which is both practical and theoretically substantial. We optimistically believe that many new
practical and theoretical approaches can be developed based on our work. Please see our reply to Reviewer #3/#5 for
more discussion on practical efficiency. We will release the code.

[Meta-Review · NeurIPS 2019]

The paper proposes an interesting technique to train neural networks while learning as well part of the architecture by splitting neurons in a principled way. More precisely, the algorithm is defining a notion of steepest descent in the space of distributions over neuron weights, which is a steepest descent in the space of these distributions equipped with the L-infty Wasserstein distance. The corresponding steepest descent algorithm retrieves the usual steepest descent direction as long as a "usual" descent direction exists, but if a local minimum is reached and some further progress could be made locally by duplicating (or make a number of other copies of) neurons and decoupling them, then the algorithm finds a locally optimal split. The paper is well written, novel, with clear simple theory. The idea is simple and elegant. The paper features a significant amount of compelling experiments with different neural network architectures both on synthetic and real data. In the discussion, the reviewers have again mentioned the above qualities of the paper. In terms of points that could potentially be improved, in addition to what is mentionned in the reviews, 1) the reviewers have expressed concern that the paper does not provide comparison of the efficacy of the proposed algorithm from the point of view of the computational cost (say if we have a lot of memory available so that memory is not constraining): is the proposed algorithm competitive in terms of running time with the baselines that consist in learning and much larger network and pruning it? 2) The reviewers have mentioned that, for networks with multiple hidden layers, it would be interesting to have some information on the architectures that are learned with the proposed algorithm. In particular if they are balanced or not. 3) That it would interesting to compare with recent more efficient pruning techniques like: Frankle, J., & Carbin, M. (2018). The Lottery Ticket Hypothesis: Finding Sparse, Trainable Neural Networks. ICLR. Please see also the updated reviews of the paper. I can only encourage the authors to take into account these comments when preparing the final version of the manuscript.